# Expected Returns and Policy Inconsistency-Aware Offline Federated Deep Reinforcement Learning

**Meng Xu** [1]  **Zhongying Chen** [2]  **Weiwei Fu** [1]  **Yan Li** [3]  **Shuguang Wang** [1]  **Jianping Wang** [1]

## Abstract

Offline Federated Deep Reinforcement Learning (FDRL) methods aggregate multiple client-side offline Deep Reinforcement Learning (DRL) models, each trained locally, to facilitate knowledge sharing while preserving privacy. Existing offline FDRL methods assign client weights during global aggregation using either simple averaging or Q-values, but they neglect the combined consideration of Q-values and policy inconsistency, the latter of which reflects the distributional discrepancy between the learned policy and the policy from offline data. This causes clients with no significant advantages in one aspect but obvious disadvantages in the other to disproportionately affect the global model, thereby degrading its capabilities in that aspect. During local training, clients in existing methods are compelled to fully adopt the global model, which negatively impacts clients when the global model is weak. To this end, we propose a novel Federated Learning (FL) framework that can be seamlessly integrated into current offline FDRL approaches to improve their performance. Our method considers both policy inconsistency and Q-values to determine the weights of client models, with the latter adjusted by a scaling factor to avoid significant numerical discrepancies with the former. The aggregated global model is then distributed to clients to facilitate their learning from the global model. The impact of the global model on the local models is reduced when a client's model performance exceeds that of the global model, thereby mitigating the influence of a weaker global model. Experiments on the Datasets for Deep Data-Driven Rein-

forcement Learning (D4RL) demonstrate that our method improves seven state-of-the-art (SOTA) offline FDRL methods across several metrics.

## 1. Introduction

Offline Deep Reinforcement Learning (DRL) enables learning a policy from static data without prolonged environmental interactions, and is widely used in areas like energy systems (Chen et al., 2024)(Chen et al., 2023b)(Chen et al., 2023a) and autonomous driving (Lee et al., 2024a). However, a single offline DRL model often faces inefficient learning due to limited data diversity, emphasizing the need for knowledge sharing among multiple models. Offline Federated Deep Reinforcement Learning (FDRL) addresses this challenge by enabling distributed training and knowledge sharing among devices or edge nodes without sharing raw data, and it has applications in robot navigation (Park & Woo, 2022) and personalized medicine (Zhou et al., 2024a).

The main challenge in offline FDRL lies in how to aggregate client models and how to use the aggregated global model to train client models. Existing offline FDRL methods can be classified into two types. The first approach uses simple averaging to aggregate local models, as seen in (Yue et al., 2024b)(Yue et al., 2024a)(Zhou et al., 2024a)(Park & Woo, 2022)(Woo et al., 2024)(Wen et al., 2023), but this method fails to prioritize higher-performing client models. The second approach assigns client weights in the global aggregation based on Q-values, as in (Rengarajan et al., 2024), where clients with higher Q-values are given more weight in the aggregation process. However, the global aggregation and local training of existing methods have the following limitations, leading to suboptimal policies. **First, existing methods fail to comprehensively consider both policy inconsistency and Q-values when calculating client importance in global aggregation**. Since offline DRL learns a policy by fitting to offline data, maximizing the expected returns of actions and ensuring the policy's fit to offline data are equally important, and both help offline DRL learn a policy with high long-term returns (Figueiredo Prudencio et al., 2024)(Levine et al., 2020). Maximizing the expected returns of actions is achieved by increasing Q-values, while

[1]Department of Computer Science, City University of Hong Kong, Hong Kong, China [2]School of Computer Science and Engineering, Southeast University, Nanjing, China [3]Pengcheng Laboratory, Shenzhen, China. Correspondence to: Jianping Wang <jianwang@cityu.edu.hk>.

*Proceedings of the $43^{rd}$ International Conference on Machine Learning*, Seoul, South Korea. PMLR 306, 2026. Copyright 2026 by the author(s).

improving the policy's fit to offline data requires minimizing policy inconsistency, which is the gap between the current policy and the potential policy represented by the offline dataset (Figueiredo Prudencio et al., 2024)(Levine et al., 2020). Thus, relying solely on Q-values for weight allocation may cause local models with no significant expected return advantage but poor data fit to occupy a larger share in the global model, severely degrading its data-fitting ability. On the other hand, focusing solely on policy inconsistency to calculate client importance neglects clients with higher expected returns, failing to effectively maximize the global model's expected return. **Second, existing methods make the local models fully adopt the global model's knowledge during local training**. Since there is no guarantee that the global model's performance will always be better than all local models, a weak global model may negatively impact stronger local models, degrading their performance.

Motivated by these observations, this work aims to propose a generic Federated Learning (FL) framework that can be seamlessly integrated into existing offline FDRL approaches to improve their performance. **To address the first limitation, we use both policy inconsistency and Q-values to determine the importance of each client**. For policy inconsistency, we use the current policy to predict actions and quantify the discrepancy between these predicted actions and those from the offline data using a distributional discrepancy metric, such as Jensen-Shannon Divergence (JSD). The Q-value is scaled by a factor derived from the reciprocal of the average absolute Q-value of a mini-batch to avoid significant numerical discrepancies with policy inconsistency when calculating client importance. Clients upload their local models and calculated importance to the server, which normalizes the importance via softmax to derive weights. The models are then aggregated with these weights to form the global model, which is sent back to the clients for local training. **To address the second limitation, we reduce the impact of a weaker global model on the local models**. During local training, each client learns from the global model. Here, we evaluate the performance of both the global and local models using the same method employed in global aggregation, which combines policy inconsistency and Q-values. If a client's performance exceeds that of the global model, the influence of the global model on the local model is reduced with a decay factor, mitigating the negative effects of a weaker global model on local models.

The key contributions are as follows:

- We introduce a novel, generic offline FDRL framework that considers both policy inconsistency and Q-values to calculate the importance of each client in global aggregation, and reduces the influence of a weak global model on stronger local models during local training.

- We provide a complexity analysis along with a theoretical evaluation to elucidate the superior performance of our approach in comparison to existing methods.

- Extensive experiments demonstrate that our method improves seven state-of-the-art (SOTA) offline FDRL methods on the Deep Data-Driven Reinforcement Learning (D4RL) dataset, achieving higher returns and D4RL scores across various FL configurations.

## 2. Related Work

**Offline DRL**. Offline DRL acquires a policy from a pre-collected, static dataset generated by a behavior policy, thereby reducing the necessity for extensive environment interactions (Figueiredo Prudencio et al., 2024)(Levine et al., 2020). A fundamental challenge in this paradigm is mitigating policy inconsistency, which requires the learned policy to minimize the discrepancy between its state-action distribution and that of the offline dataset (Figueiredo Prudencio et al., 2024)(Levine et al., 2020). To address this issue, various strategies have been proposed, including regularization techniques (Fujimoto & Gu, 2021), data rebalancing approaches (Jiang et al., 2023)(Yu et al., 2022)(Hong et al., 2023), and weighted behavior cloning methods (Peng et al., 2023)(Liu et al., 2024). Nevertheless, conventional offline DRL remains constrained to training solely on the fixed dataset, which inherently limits the model's capacity to learn beyond the scope of that data.

**FL**. FL is a distributed machine learning approach that allows clients to collaboratively train a shared model without exchanging data, thereby ensuring privacy. FedAvg is one example that facilitates this collaboration without data sharing (McMahan et al., 2017)(Lee et al., 2024b). A significant challenge in FL is data heterogeneity, which can slow down and destabilize convergence (Karimireddy et al., 2020)(Wang et al., 2024)(Ahmed et al., 2024). To address this, the community has developed various methods to calculate client importance for effective global aggregation, including approaches based on dataset size proportions, as seen in methods like FedProx (Karimireddy et al., 2020). Other strategies utilize local test performance as a reputation score for local models to determine aggregation weights, as mentioned in (Wang & Kantarci, 2021), or assess the importance of local models based on local Shapley values, as discussed in (Tang et al., 2021). Recently, FL has also been applied to fine-tune large models (Wu et al., 2024)(Liu et al., 2025)(Ye et al., 2024). While FL primarily focuses on supervised learning with local client data, offline FDRL aims to train a policy using offline samples, targeting a policy that outperforms the one represented by those samples. This necessitates the development of aggregation strategies for offline FDRL that differ from traditional FL.

**FDRL**. FDRL trains DRL models through knowledge sharing while ensuring privacy. Previous research has focused on online FDRL, which includes Horizontal FDRL (HFDRL) and Vertical FDRL (VFDRL). HFDRL involves independent agents operating in different environments (Cha et al., 2020)(Jiang et al., 2025)(Wang et al., 2023), whereas VF-DRL emphasizes collaboration within a shared environment with limited observations (Zhuo et al., 2019). Methods for calculating client importance in online FDRL include average weighting and strategy KL divergence weighting, such as FedKL (Xie & Song, 2023). Due to inefficiencies associated with prolonged online interactions, offline FDRL has emerged, which trains multiple offline client DRL models using local static datasets. Existing offline FDRL methods can be categorized into two groups: one that uses simple averaging to aggregate client models (Yue et al., 2024b)(Yue et al., 2024a)(Zhou et al., 2024a)(Park & Woo, 2022)(Woo et al., 2024)(Wen et al., 2023)(Qiao et al., 2025), thereby ignoring client heterogeneity, and another that employs Q-values (Rengarajan et al., 2024) to calculate client weights. However, current offline FDRL methods face two limitations. First, they do not comprehensively consider both policy inconsistency and Q-values when determining client importance, which can lead to clients lacking significant advantages in one aspect while suffering clear disadvantages in another, thus impairing the global model in that area. Second, these methods require clients to fully adopt the global model, which can harm stronger clients when the global model is suboptimal.

## 3. Preliminaries

**FL**. FL aggregates various client models into a global model $\theta$ without sharing local device data. The aggregated model is then redistributed to each client for local training. Let $N_a$ be the total number of clients, with each client operating a local model. Let $\mathcal{L}_i$ represent the loss over the local data for the $i$-th client model $\theta_i$, and $w_i$ represent the weight assigned to the $i$-th client. The objective of FL is to minimize the loss function $\mathcal{L}$, which is the weighted aggregation of individual clients' losses. This can be expressed as: $\min_\theta \mathcal{L} = \min \sum_{i=1}^{N_a} w_i \mathcal{L}_i(\theta_i)$. The aggregation method, including how $w_i$ is computed, is crucial in FL. For example, $w_i$ can be determined by the ratio of $n_i$ (the sample batch size for the $i$-th client) to the total sum of all $n_i$ values. Recently, FL has advanced model aggregation techniques, including FedProx (Yuan & Li, 2022), FedCAda (Zhou et al., 2024b), and FedAdam (Ju et al., 2024).

**Offline DRL**. DRL is a learning method based on the Markov Decision Process (MDP), defined as $\mathcal{M} = \langle \mathcal{S}, \mathcal{A}, r, \mathbb{P}, \gamma \rangle$, which includes the state space $\mathcal{S}$, action space $\mathcal{A}$, reward function $r$, transition dynamics $\mathbb{P}$, and discount factor $\gamma$. Unlike conventional DRL, which obtains

samples through interaction with the environment, offline DRL aims to learn a policy $\pi$ using a static dataset $\mathcal{D}$ that contains transitions $(s, a, r, s')$ without further interaction. Each tuple $(s, a, r, s')$ in $\mathcal{D}$ represents the state, action, reward, and next state, collected by a behavior policy $\pi_b$. The goal of offline DRL is to learn a policy $\pi(s)$ that maximizes the long-term reward $J(\pi(s))$ over the static dataset: $J(\pi(s)) = \max \mathbb{E}_{\pi(s)} \left[ \sum_{i=t}^{T} \gamma^{i-t} r_i \mid s_0, a_0 \right]$ where $T$ is the learning duration, and $t$ starts at 0. In offline DRL, since the dataset $\mathcal{D}$ is pre-collected through another policy, the agent samples a mini-batch $\mathcal{D}_0$ from $\mathcal{D}$ at each step to update the model. This mini-batch has a different distribution from the current actor being updated, represented by the difference between the actor's predicted actions $\hat{a} = \pi(s; \theta^\mu)$ and the offline actions $a$. This difference, referred to as policy inconsistency, is minimized in offline DRL. In DRL, the actor-critic method is a widely used framework comprising two key components. The actor, denoted as $\theta^\mu$, parameterizes the policy $\pi(s; \theta^\mu)$, while the critic, represented by $\theta$, parameterizes the Q function $Q(s, a; \theta)$. The Q function estimates the expected cumulative reward when following the policy $\pi$, starting from state $s$ and taking action $a$.

**Offline FDRL**. The offline FDRL implementation operates in a distributed offline setting, where $N_a$ agents collaboratively develop a policy under the guidance of a central server, without sharing raw trajectories or interacting with the environment. Each agent $i \in \{1, \ldots, N_a\}$ maintains a local dataset $\mathcal{D}_i \doteq \left\{ \left( s_{i,j}, a_{i,j}, r_{i,j}, s'_{i,j} \right) \right\}_{j=1}^{D_i}$, containing transition tuples generated by an unknown policy. The objective is to discover an optimal policy by leveraging the distributed datasets $\{\mathcal{D}_i\}_{i=1}^{N_a}$, maximizing the long-term reward. To achieve this, offline FDRL allows each client to train a local model $\theta_i$ for a certain number of epochs using its local dataset $\mathcal{D}_i$. The server then aggregates the models from all participating clients, weighted accordingly, to obtain a global model $\theta$, which is represented as $\theta = \sum_{i=1}^{N_a} w_i \theta_i$. This global model is sent back to the clients for continued training. The process of local training followed by global aggregation is repeated until the global model is fully trained. The performance of various offline FDRL methods is assessed based on the cumulative reward from the global model, with higher values reflecting better performance.

Existing offline FDRL methods have two limitations. 1) **Inefficient global aggregation methods**. One group of methods (Yue et al., 2024b)(Yue et al., 2024a)(Zhou et al., 2024a)(Park & Woo, 2022)(Woo et al., 2024)(Wen et al., 2023) calculates client weights $w_i$ by averaging (e.g., $w_i = \frac{1}{N_a}$), which overlooks the fact that heterogeneous clients should be assigned different weights. Another approach (Rengarajan et al., 2024) assigns client weights based on Q-value size, for example $w_i \propto Q_i$, but it neglects policy inconsistency. This is likely to cause clients with no signif-

icant advantages in Q-values but poor data-fitting abilities to occupy a larger share in global aggregation and severely degrade their data-fitting ability. 2) **Inefficient local training methods**. During local training, existing methods force clients to fully adopt the global model's experience, causing a weak global model to negatively impact local models.

## 4. Methodology

This section outlines the framework and details, followed by a complexity analysis of our method. **The appendix includes pseudocode, a comprehensive complexity analysis, and a detailed theoretical analysis of our approach.**

### 4.1. The Framework for Our Method

Fig. 1 (with $N$ clients as an example) illustrates the framework of our method, which encompasses both **local training** and **global aggregation**. During the global aggregation phase, each client samples a mini-batch from its local dataset and computes the policy inconsistency using both the actions predicted by the policy and the offline actions. This is combined with the Q-value to determine each client's importance, which, along with the local model, is uploaded to the server. The server computes the weight of each client using softmax normalization based on importance and performs a weighted sum of their models to obtain the global model. In the local training phase, the global model is sent to each client, where it learns from the global model using the approach directly adopted from our integrated baseline FDRL method. A decay factor is applied to reduce the impact of the global model on local updates when the global model's performance, as evaluated through policy inconsistency and Q-value, is weaker than the local model's performance. Once local training is complete, global aggregation occurs again, repeating until training concludes.

### 4.2. Global Aggregation

After local training on each client for a specified number of epochs, the models are uploaded to the server for aggregation. Let $Q_{\theta^{Q_i}}(s, a)$ represent the Q-value for the $i$-th client, typically generated by the first critic in the local offline DRL model, $\theta_i^\mu$ the actor for the $i$-th client, and $a_i$ the action from the $i$-th client's mini-batch $\mathcal{D}_{0i}$, which is sampled from the static dataset $\mathcal{D}_i$. The importance assessment for each client takes into account both the Q-value and the policy inconsistency $Dis(\theta_i^\mu(s), a_i)$. The importance $I_i$ can be calculated using any combination of Q-value and $Dis(\theta_i^\mu(s), a_i)$, such as Q-value minus $Dis(\theta_i^\mu(s), a_i)$. The guiding principle is that a smaller $Dis(\theta_i^\mu(s), a_i)$ is preferable, while a larger Q-value is desirable. Here, we present a straightforward calculation for the importance $I_i$ for the $i$-th client, as follows:

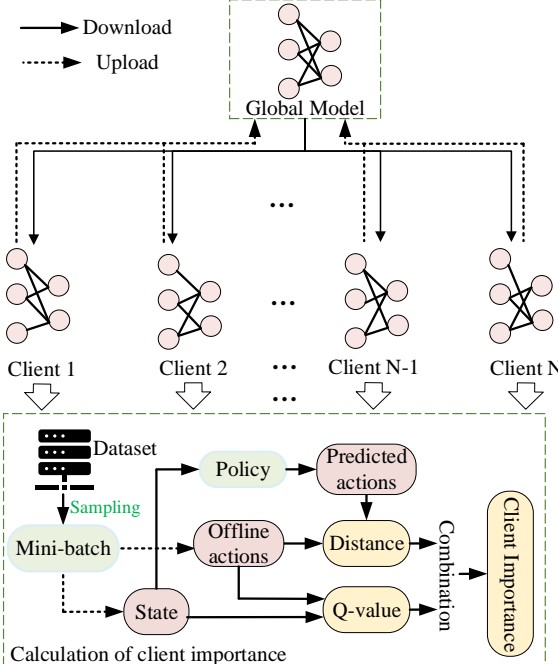

*Figure 1.* The framework of our approach.

$$I_i = \mathbb{E}_{(s,a) \sim \mathcal{D}_{0i}} \left[ \kappa_i Q_{\theta^{Q_i}}(s, a) - Dis(\theta_i^\mu(s), a_i) \right] \quad (1)$$

Here, $Dis(\theta_i^\mu(s), a_i)$ can be computed using any distributional difference metric, such as the squared difference: $Dis(\theta_i^\mu(s), a_i) = (\pi_{\theta_i^\mu}(s) - a_i)^2$. In fact, in addition to this method for calculating $I_i$, our approach is also compatible with various combinations of $Q_{\theta^{Q_i}}(s, a)$ and $Dis(\theta_i^\mu(s), a_i)$, such as dividing the two factors. Different calculation methods for $I_i$ will yield varying performance outcomes. The server aggregates local models as follows.

**(1) Calculating Policy Inconsistency**: Besides the basic squared difference, advanced distribution discrepancy measures such as the JSD, a robust method for assessing distribution differences and resistant to outliers, can be used. Specifically, we first compute the difference between the predicted and offline actions, $\pi_{\theta_i^\mu}(s) - a_i$, and map this difference to a multivariate Gaussian distribution $\mathcal{N}(\mu, \Sigma)$. The multivariate Gaussian is ideal for modeling complex action distributions in DRL within high-dimensional spaces (Williams, 1992) (Todorov et al., 2012) (Nasiriany et al., 2021) (Hollenstein et al., 2022). The mean and covariance $(\mu, \Sigma)$ are given by:

$$\mu = \frac{1}{|\mathcal{D}_{0i}|} \sum_{(s,a) \in \mathcal{D}_{0i}} (\pi_{\theta_i^\mu}(s) - a_i) \quad (2)$$

$$\Sigma = \frac{\sum_{(s,a) \in \mathcal{D}_{0i}} \left(\pi_{\theta_i^\mu}(s) - a_i - \mu\right)^\top \left(\pi_{\theta_i^\mu}(s) - a_i - \mu\right)}{|\mathcal{D}_{0i}| - 1} \quad (3)$$

Next, we compute the JSD to evaluate $Dis(\theta_i^\mu(s), a_i)$ between the distribution $\mathcal{N}(\mu, \Sigma)$ and the standard multivariate Gaussian distribution $\mathcal{N}(\mathbf{0}, \sigma\mathbf{I})$, where $\mathbf{I}$ is the identity matrix and $\sigma = 0.15$ is a commonly used value, as follows:

$$
\begin{aligned}
Dis(\theta_i^\mu(s), a_i) &= \mathrm{JSD}\left(\mathcal{N}(\mu, \Sigma)|\mathcal{N}(\mathbf{0}, \sigma\mathbf{I})\right) \\
&= \frac{1}{2}\mathrm{KLD}\left(\mathcal{N}(\mu, \Sigma)\|\frac{\mathcal{N}(\mathbf{0}, \sigma\mathbf{I}) + \mathcal{N}(\mu, \Sigma)}{2}\right) + \\
&\quad \frac{1}{2}\mathrm{KLD}\left(\mathcal{N}(\mathbf{0}, \sigma\mathbf{I})\|\frac{\mathcal{N}(\mathbf{0}, \sigma\mathbf{I}) + \mathcal{N}(\mu, \Sigma)}{2}\right)
\end{aligned} \quad (4)
$$

where $\mathrm{JSD}(\cdot\|\cdot)$ denotes the JSD and $\mathrm{KLD}(\cdot\|\cdot)$ represents the Kullback-Leibler Divergence (KLD).

**(2) Calculating Client Importance**: Here, we mainly calculate $\kappa_i$, which balances $(\pi_{\theta_i^\mu}(s) - a_i)^2$ and $Q_{\theta^{Q_i}}(s, a)$, and is given by $\kappa_i = \frac{1}{\frac{1}{|\mathcal{D}_{0i}|}\sum_{(s_i, a_i)\sim\mathcal{D}_{0i}}|Q(s_i, a_i)|}$. The term $\kappa_i$ is essential because each client clips actions to the range of [-1, 1], ensuring that $Dis(\theta_i^\mu(s), a_i)$ does not become excessively large, which could lead to significant numerical differences from $Q_{\theta^{Q_i}}(s, a)$. For instance, when using the squared difference, $(\pi_{\theta_i^\mu}(s) - a_i)^2$, $Dis(\theta_i^\mu(s), a_i)$ can be at most 4. Similarly, when employing JSD, $Dis(\theta_i^\mu(s), a_i)$ has a manageable upper limit, such as 0.69. In contrast, $Q_{\theta^{Q_i}}(s, a)$ is positively correlated with the scale of rewards, which theoretically has no upper bound. Therefore, we use the inverse of the average absolute Q-values over a mini-batch to scale $Q_{\theta^{Q_i}}(s, a)$ and avoid excessive numerical differences with $Dis(\theta_i^\mu(s), a_i)$. Then, we obtain the importance vector for the clients, $\mathbf{I} = (I_1, I_2, ..., I_{N_a})$.

**(3) Weighted Aggregation**: We normalize the importance values of $\mathbf{I}$ using the softmax operation, as each client's importance $I_i$ may differ in magnitude. The normalized values of $\mathbf{I}$ are then used as weights, $\mathbf{w}$, as follows

$$
\begin{aligned}
\mathbf{w} = (w_1, w_2, ..., w_{N_a}) &= \mathrm{softmax}(I_1, I_2, ..., I_{N_a}) \\
&= \left(\frac{e^{I_1}}{\sum_{i=1}^{N_a} e^{I_i}}, \frac{e^{I_2}}{\sum_{i=1}^{N_a} e^{I_i}}, ..., \frac{e^{I_{N_a}}}{\sum_{i=1}^{N_a} e^{I_i}}\right)
\end{aligned} \quad (5)
$$

The resulting weight $w_i$ is positively correlated with $I_i$. A higher $I_i$ indicates greater importance of the client, thus receiving a larger weight $w_i$ in the global aggregation. Finally, we compute the aggregated global model $\theta_{global}^\mu, \theta_{global}^Q$ at the server using a weighted sum of the clients' actors and critics: $\theta_{global}^\mu = \sum_{i=1}^{N_a} w_i\theta_i^\mu; \theta_{global}^Q = \sum_{i=1}^{N_a} w_i\theta_i^Q$.

### 4.3. Local Training

After global aggregation, the global critic $\theta_{global}^Q$ and global actor $\theta_{global}^\mu$ are obtained. These global models are then downloaded by the clients, who aim to learn from them using their respective loss functions. Since the actors and critics of the clients have different loss functions, let $\mathcal{L}_{local}(\theta_i^Q)$

represent the critic loss for client $i$, and let $\mathcal{L}_{local}(\theta_i^\mu)$ represent the actor loss for client $i$. Given that local models may outperform the global model, each client (e.g., client $i$) assesses both the performance of the global model, denoted as $I_{glo}$, and its own performance, denoted as $I_i$, by randomly sampling a mini-batch prior to each local update.

If $I_i > I_{glo}$, a decay factor $\beta_i$ is applied to reduce the influence of the client's loss on the model update (e.g., for client $i$). The decay factor is updated as $\beta_i^t = \beta_i^{t-1} * \zeta$, with $\beta_i$ initially set to 1 and $\zeta$ set to 0.99. Here, $t$ represents the current iteration of the local update, during which the global model's performance is weaker than that of client $i$. As the global model increasingly underperforms in more local updates, its influence on the local model continues to diminish, as indicated by $\beta_i^{t-1} * \zeta$, which essentially corresponds to a stronger penalty for the weaker global model. Conversely, if $I_i \leq I_{glo}$, the decay factor $\beta_i$ for this local update is set to 1. Each local model's critic and actor updates are as follows:

**(1) Updating Critics**: The loss for the $j$-th critic $\theta_i^{Q_j}$ of the $i$-th client is given by

$$
\theta_i^{Q_j} = \arg\min_{\theta^Q}\mathcal{L}(\theta_i^{Q_j}); \mathcal{L}(\theta_i^{Q_j}) = \beta_i\mathcal{L}_{local}(\theta_i^Q) \quad (6)
$$

where $\mathcal{L}_{local}(\theta_i^Q)$ denotes the baseline offline FDRL method's original critic loss.

**(2) Updating Actor**: Next, we update the clients' actors. Let $\mathcal{L}_{local}(\theta_i^\mu)$ denote the original loss for the $i$-th client's actor $\theta_i^\mu$ in the baseline offline FDRL method. The final loss for the $i$-th client's actor, $\mathcal{L}(\theta_i^\mu)$, is given by

$$
\pi_{\theta_i^\mu} = \arg\min_\pi\mathcal{L}(\theta_i^\mu); \mathcal{L}(\theta_i^\mu) = \beta_i\mathcal{L}_{local}(\theta_i^\mu) \quad (7)
$$

### 4.4. Complexity Analysis

Our method has a time complexity of $O(N_a + N_a \cdot |\mathcal{D}_{0i}|)$, introducing minimal computational overhead to existing offline FDRL methods. Since current offline FDRL methods upload client weights and local models to the server, our approach does not add new model components during local training or alter the components uploaded by clients to the server. As a result, the space complexity and communication cost of our method remain consistent with those of current offline FDRL methods.

### 4.5. Theoretical Analysis

This section presents two theorems, with Theorem 1 demonstrating the differences in long-term rewards between two policies. From Theorem 1, we derive Theorem 2, which illustrates how our method enhances existing approaches by incorporating policy inconsistency into global aggregation.

**Theorem 1**: Let $J(\pi^A(s))$ and $J(\pi^B(s))$ represent the long-term rewards of two policies, $\pi^A(s)$ and $\pi^B(s)$. Therefore, the difference between $J(\pi^A(s))$ and $J(\pi^B(s))$ can be bounded as follows:

$$|J(\pi^A(s)) - J(\pi^B(s))| \leq \frac{R_M K_\pi}{1-\gamma} \max_{s \in \mathcal{S}} \|\pi^A(s) - \pi^B(s)\|$$
(8)

This derivation demonstrates that the difference in long-term cumulative rewards $J$ between two policies is proportional to their discrepancy, with $\frac{R_M K_\pi}{1-\gamma}$ serving as a positive constant applicable to all policies. Here, $R_M$ represents the upper bound of reward magnitude, and $K_\pi$ denotes the Lipschitz constant.

**Theorem 2**: Let $\pi_k(s)$ be the policy of the $k$-th local model, assuming it has the minimum policy inconsistency with its corresponding offline dataset policy $\pi_k^{off}(s)$ among all clients. The global model's policy is represented as $\pi(s) = \sum_{k=1}^{N_a} w_k \pi_k(s)$, while $\pi^*(s)$ denotes the theoretical optimal global policy. The difference between the optimal policy $J(\pi^*(s))$ and the policy learned by various offline FDRL methods $J(\pi(s))$ can be bounded as follows:

$$|J(\pi^*(s)) - J(\pi(s))| \leq \frac{R_M K_\pi}{1-\gamma} \left( \max_{s \in \mathcal{S}} \|\pi^*(s) - \pi_k^{off}(s)\| \right.$$

$$\left. + \|\pi_k^{off}(s) - \pi_k(s)\| + \|\pi_k(s) - \pi(s)\| \right)$$
(9)

**Discussion**: In this derivation, the terms $\max_{s \in \mathcal{S}} \|\pi^*(s) - \pi_k^{off}(s)\| + \max_{s \in \mathcal{S}} \|\pi_k^{off}(s) - \pi_k(s)\|$ remain constant across different offline FDRL methods. Our approach gives greater weight $w_k$ to models from clients with minimal policy inconsistency during global aggregation, resulting in a smaller difference between $\pi(s) = \sum_{k=1}^{N_a} w_k \pi_k(s)$ and $\pi_k(s)$. Consequently, $\max_{s \in \mathcal{S}} \|\pi_k(s) - \pi(s)\|$ is reduced compared to existing offline FDRL methods. This leads to a tighter upper bound on the performance difference between our method and the theoretically optimal global policy, allowing our method to more closely approach the optimal global policy. Therefore, by incorporating policy inconsistency into global aggregation, our approach improves current offline FDRL methods.

# 5. Experiment and Analysis

This section details the experimental settings, results, and analysis. The software stack includes Torch 1.2.0, Gym 0.16.0, and mujoco-py 1.50.0.1, whereas the hardware configuration comprises an Intel Core i7-9700 processor, 64 GB RAM, and an NVIDIA RTX 2080 GPU. Each method is tested with five random seeds. **The appendix presents the implementation details of our method.**

## 5.1. Experiment Settings

**(1) Baselines**. Seven SOTA offline FDRL methods: **Federated Diffusion Q-Learning (FDQL)** (Wen et al., 2023), **Federated DRL with Dual Regularization (FDRLDR)** (Yue et al., 2024b), **Federated Offline Reinforcement Learning (FORL)** (Yue et al., 2024a), **Collaborative Single-policy Coverage (CSC)** (Woo et al., 2024), **Offline Federated Reinforcement Learning with Mixed-Quality Data (FOVA)** (Qiao et al., 2025), **Model-contrastive federated learning (MOON)** (Li et al., 2021), and **Federated Ensemble-Directed Offline Reinforcement Learning Algorithm (FEDORA)** (Rengarajan et al., 2024).

**(2) Benchmarks**. We use the D4RL dataset (Fu et al., 2020) as a benchmark, which includes four offline MuJoCo tasks: HalfCheetah, Hopper, Walker2d, and Ant. Our setup involves 20 clients, each with a local dataset of size $|\mathcal{D}_i| = 5000$. The global model is evaluated on the expert dataset for the four MuJoCo tasks. To simulate real-world scenarios, we: 1) **Different Datasets for Clients**: 10 clients' data are sampled randomly without replacement from the D4RL expert dataset, while the other 10 clients' data are sampled randomly without replacement from the D4RL medium dataset; 2) **Clients Unaware of Dataset Quality**: Both clients and the server are unaware of dataset quality, with no access to the environment; 3) **Random Client Participation**: In each FL round $t$, we randomly select $N_a = 10$ clients to participate in the FL process. Each client performs $T = 20$ epochs of local training per round, equating to about 380 local gradient steps.

Furthermore, to comprehensively evaluate the performance of our method, we set up six additional FL configurations. **First**, we explore federation with varying proportions of medium client participants, adjusting the proportion of clients using the medium dataset to 25% and 75%. **Second**, we examine different numbers of local training epochs by changing the local training duration after each global aggregation to 10 and 30 epochs. **Third**, we maintain a fixed proportion of 50% aggregation participating clients while varying both the total number of clients and those participating in global aggregation to 30:15 and 40:20. **Fourth**, we compare the performance of different methods with client local datasets of sizes $|\mathcal{D}_i|$ set to 2500 and 10000. **Fifth**, we analyze different proportions of aggregation participants, keeping the total number of clients at 20 and setting participation ratios to 5:20 and 15:20. **Sixth**, we set up 30 clients, with 15 participating in FL; 10 clients have the expert dataset, 10 have the medium dataset, and another 10 have a random dataset, allowing us to analyze the impact of introducing lower-quality datasets on our method.

**(3) Metrics**. We evaluate different methods using four metrics: **the final D4RL score** (presented in the main text), **the final episode return** (presented in the appendix), **the**

*Table 1.* Comparison with SOTA methods in terms of final D4RL score (Mean ± Standard deviation).

| Methods | HalfCheetah | Hopper |
|---|---|---|
| FDQL | $42.78 \pm 6.25$ | $44.38 \pm 9.39$ |
| **Ours+FDQL** | $\mathbf{48.55} \pm 9.51$ | $\mathbf{50.43} \pm 13.59$ |
| FDRLDR | $48.3 \pm 4.30$ | $44.44 \pm 14.63$ |
| **Ours+FDRLDR** | $\mathbf{49.28} \pm 5.82$ | $43.11 \pm 10.34$ |
| FORL | $41.0 \pm 8.72$ | $46.92 \pm 10.74$ |
| **Ours+FORL** | $\mathbf{51.74} \pm 8.36$ | $\mathbf{53.35} \pm 19.75$ |
| FEDORA | $26.75 \pm 18.36$ | $49.44 \pm 11.72$ |
| **Ours+FEDORA** | $\mathbf{48.61} \pm 5.65$ | $\mathbf{84.44} \pm 35.85$ |
| CSC | $42.45 \pm 14.16$ | $42.15 \pm 10.20$ |
| **Ours+CSC** | $\mathbf{46.37} \pm 7.72$ | $\mathbf{97.07} \pm 14.35$ |
| FOVA | $46.99 \pm 6.76$ | $41.71 \pm 9.96$ |
| **Ours+FOVA** | $42.35 \pm 5.81$ | $\mathbf{48.53} \pm 10.71$ |
| MOON | $33.45 \pm 5.95$ | $48.40 \pm 17.27$ |
| **Ours+MOON** | $\mathbf{44.84} \pm 10.46$ | $\mathbf{60.24} \pm 10.94$ |

| Methods | Walker2d | Ant |
|---|---|---|
| FDQL | $68.78 \pm 20.66$ | $79.77 \pm 34.55$ |
| **Ours+FDQL** | $\mathbf{77.66} \pm 13.36$ | $\mathbf{82.1} \pm 32.27$ |
| FDRLDR | $74.18 \pm 14.19$ | $65.48 \pm 22.65$ |
| **Ours+FDRLDR** | $\mathbf{75.51} \pm 15.25$ | $\mathbf{77.23} \pm 25.89$ |
| FORL | $63.03 \pm 13.47$ | $59.19 \pm 31.17$ |
| **Ours+FORL** | $\mathbf{71.69} \pm 20.00$ | $\mathbf{71.17} \pm 26.76$ |
| FEDORA | $67.99 \pm 18.30$ | $44.3 \pm 22.14$ |
| **Ours+FEDORA** | $\mathbf{109.29} \pm 0.36$ | $\mathbf{61.67} \pm 28.83$ |
| CSC | $108.56 \pm 0.82$ | $62.58 \pm 15.84$ |
| **Ours+CSC** | $65.12 \pm 15.33$ | $\mathbf{67.16} \pm 27.13$ |
| FOVA | $59.75 \pm 11.23$ | $64.75 \pm 26.00$ |
| **Ours+FOVA** | $\mathbf{71.42} \pm 24.39$ | $\mathbf{73.75} \pm 31.74$ |
| MOON | $65.14 \pm 17.46$ | $15.59 \pm 5.31$ |
| **Ours+MOON** | $\mathbf{69.13} \pm 16.42$ | $\mathbf{80.90} \pm 12.02$ |

**episode return** (presented in the appendix), and **D4RL score** (presented in the appendix). **The episode return** is the average reward per communication round, typically displayed as a curve showing its progression. The **D4RL score** is a metric used to evaluate offline DRL performance, utilizing the normalized reward (Fu et al., 2020) and displayed as a curve showing training progress. **The final D4RL score** is the average D4RL score from the global model over the last 10 communication rounds, while **the final episode return** is the average episode return over the same period. Higher values in these metrics are favored.

### 5.2. Comparison with SOTA Offline FDRL methods

This section presents our improvements to seven SOTA methods across four D4RL tasks: HalfCheetah, Hopper, Walker2d, and Ant. These improvements are evaluated based on the final D4RL score, which is presented in Table 1. The results indicate that after integrating our method, which replaces the existing global aggregation methods and incorporates our decay factor into local training, the performance of existing offline FDRL methods improves in the vast majority of cases, resulting in higher episode returns and D4RL scores. This demonstrates that our framework enhances the performance of existing methods. Among the seven baseline methods, FDRLDR, FDQL, CSC, MOON, FOVA, and FORL aggregate client models by averaging with equal weight. However, due to client heterogeneity, these approaches fail to prioritize higher-performing mod-

els. Similarly, FEDORA computes weights based solely on the Q-values of client models, ignoring the impact of policy inconsistency on offline FDRL performance. This may cause clients with no significant advantages in Q-values but poor data-fitting abilities to obtain higher weights in global aggregation, which severely degrades the global model's data-fitting ability and fails to fully maximize the expected returns of its policy. Meanwhile, existing methods require local models to fully adopt the global model, leading to weak global models negatively impacting local models. In contrast, our method integrates policy inconsistency and Q-values to compute client weights, thereby amplifying the impact of client models with better overall performance on the global model. Additionally, our method assesses both the global and local models' capabilities, mitigating the negative impact of a weak global model on local models.

Additionally, when training the Ours+FORL and Ours+FEDORA methods on the HalfCheetah task, we found that $Dis(\theta_i^\mu(s), a_i)$ varied around 0.2, while $\kappa_i Q_{\theta Q_i}(s, a)$ varied around 2. This suggests that the numerical difference between the two is not significant, underscoring the need to use $\kappa$ to scale $Q_{\theta Q_i}(s, a)$. Finally, we evaluated the hardware efficiency of our method by testing the average Floating Point Operations (FLOPs) and average Video Random Access Memory (VRAM) usage in megabytes (MB) for all clients during each local training epoch. FLOPs measure computational workload, while VRAM usage assesses memory consumption. Our tests indicate that the FDRLDR method and our method, denoted as Ours+FDRLDR, exhibit similar VRAM and FLOPs, both approximately 88.99 MB and 4.11e+06, respectively, for the HalfCheetah environment. In the case of the Hopper environment, the VRAM and FLOPs are around 86.15 MB and 3.98e+06. Therefore, our method does not impose a significant additional computational burden.

### 5.3. Ablation and Hyperparameter Sensitivity Analysis

This section first presents an ablation study demonstrating that 1) assessing client model importance solely based on policy inconsistency, referred to as Global Aggregation Using Policy Inconsistency (GAPI), 2) assessing client model importance solely based on Q-value, referred to as Global Aggregation Using Q-value (GAQ), and 3) not employing a decay strategy for the influence of the global model on local models, referred to as Our Approach Without Decay (OWD) ($\zeta = 1$), are all suboptimal. The first two emphasize the need to use both policy inconsistency and Q-value simultaneously to evaluate client model importance, while the latter highlights the necessity of applying a decay strategy. We then perform a sensitivity analysis on the hyperparameter $\zeta$ in our method. Two baselines are set up: 1) Ours (0.8), where the decay strategy is applied with $\zeta = 0.8$; and 2) Ours (0.9), where the decay strategy is applied with $\zeta = 0.9$.

We use four SOTA methods as baselines, with HalfCheetah as the validation task. The experimental results, presented in Table 2, are based on final D4RL scores.

**First, we analyze the ablation study**, specifically comparing the integration of GAPI into existing methods (GAPI+baseline), the integration of GAQ into existing methods (GAQ+baseline), the integration of OWD into existing methods (OWD+baseline), and our approach (Ours+baseline). The results show that GAPI+baseline and GAQ+baseline produce weaker D4RL scores compared to our approach. This is because relying solely on policy inconsistency does not identify clients with higher expected returns, represented by Q-values, which prevents the global model from effectively maximizing the policy's expected returns and leads to suboptimal performance. Similarly, relying solely on Q-values fails to identify clients with lower policy inconsistency, resulting in suboptimal policies. Second, without a decay strategy, as seen in the OWD+baseline method, the weaker global model negatively impacts the local models. Consequently, the OWD+baseline method is surpassed by the Ours+baseline method. This strategy helps mitigate the issue by enhancing local model performance, which in turn improves the global model. Additionally, since the OWD+baseline method only integrates our global aggregation strategy into baseline methods by replacing their global aggregation strategies, it surpasses the baseline methods in most cases. This indicates that our global aggregation strategy outperforms existing offline FDRL methods' global aggregation strategies.

**Next, we analyze the hyperparameter sensitivity**. The results show that Ours (0.8) and Ours (0.9) perform worse compared to our method ($\zeta = 0.99$), confirming the optimality of our current hyperparameter settings. If the decay rate is too large, as in Ours (0.8) and Ours (0.9), performance deteriorates. This is because, during certain local updates, the global model may be weaker than the client model $i$, and a high decay level (e.g., $\zeta = 0.9$) reduces the global model's negative impact by adjusting $\beta_i$ to 0.9 (from an initial value of 1). If the global model weakens further, $\beta_i$ decreases further (e.g., to 0.81), drastically reducing the global model's influence on the local model. However, due to the inherent instability of DRL model performance (Xu et al., 2025b)(Xu et al., 2025a), where the global model may be weaker than the local model during some updates, this does not imply that the global model always provides negative experiences or consistently performs worse than the local model. Therefore, an excessively large decay level reduces the potential for the global model to positively influence local model updates, leading to a suboptimal global model. Based on these results, the decay level should not be too large (e.g., using a very small $\zeta$).

*Table 2.* Ablation study and hyperparameter sensitivity analysis.

| Methods | FDQL | FDRLDR |
|---|---|---|
| baseline | $42.78 \pm 6.25$ | $48.3 \pm 4.3$ |
| GAPI+baseline | $45.64 \pm 8.49$ | $48.8 \pm 6.27$ |
| GAQ+baseline | $45.9 \pm 3.2$ | $42.6 \pm 5.8$ |
| OWD+baseline | $45.12 \pm 8.87$ | $45.05 \pm 6.14$ |
| Ours (0.8)+baseline | $45.32 \pm 9.13$ | $48.47 \pm 5.04$ |
| Ours (0.9)+baseline | $47.8 \pm 7.03$ | $47.78 \pm 7.32$ |
| **Ours+baseline** | $\mathbf{48.55} \pm 9.51$ | $\mathbf{49.28} \pm 5.82$ |

| Methods | FORL | FEDORA |
|---|---|---|
| baseline | $41.0 \pm 8.72$ | $26.75 \pm 18.36$ |
| GAPI+baseline | $50.27 \pm 6.65$ | $46.38 \pm 6.32$ |
| GAQ+baseline | $43.4 \pm 10.4$ | $47.00 \pm 7.5$ |
| OWD+baseline | $48.02 \pm 6.62$ | $48.53 \pm 6.39$ |
| Ours (0.8)+baseline | $46.84 \pm 4.57$ | $47.29 \pm 4.82$ |
| Ours (0.9)+baseline | $44.34 \pm 6.9$ | $45.11 \pm 6.57$ |
| **Ours+baseline** | $\mathbf{51.74} \pm 8.36$ | $\mathbf{48.61} \pm 5.65$ |

*Table 3.* Integration with different components

| Methods | FDQL | FDRLDR |
|---|---|---|
| SD+baseline | $45.66 \pm 5.41$ | $47.42 \pm 5.73$ |
| KLD+baseline | $43.92 \pm 6.51$ | $45.29 \pm 8.31$ |
| Method II+baseline | $13.49 \pm 14.52$ | $1.91 \pm 3.98$ |
| Method III+baseline | $41.89 \pm 2.18$ | $41.58 \pm 2.53$ |
| **JSD+baseline (Ours)** | $\mathbf{48.55} \pm 9.51$ | $\mathbf{49.28} \pm 5.82$ |

| Methods | FORL | FEDORA |
|---|---|---|
| SD+baseline | $49.2 \pm 8.29$ | $45.82 \pm 5.83$ |
| KLD+baseline | $49.76 \pm 5.27$ | $44.59 \pm 6.9$ |
| Method II+baseline | $2.66 \pm 5.64$ | $4.67 \pm 6.33$ |
| Method III+baseline | $41.9 \pm 2.54$ | $42.08 \pm 2.80$ |
| **JSD+baseline (Ours)** | $\mathbf{51.74} \pm 8.36$ | $\mathbf{48.61} \pm 5.65$ |

## 5.4. Integration with Different Components

This section evaluates our method's performance when integrated with different components, with experimental results presented in Table 3 in terms of the final D4RL score, using HalfCheetah as the validation task.

**First**, we assess how our method's performance changes based on various metrics for evaluating policy inconsistency, focusing on three indicators: **Squared Difference (SD)**, **KLD**, and **JSD**. Our results indicate that optimal performance is achieved when employing JSD to measure policy inconsistency. Unlike SD, which relies on Euclidean geometry, JSD uses a multivariate Gaussian distribution to capture differences among actions, making it particularly suitable for high-dimensional action spaces typical in DRL tasks, such as those in MuJoCo. Furthermore, JSD offers a more comprehensive approach than KLD, as it mitigates the asymmetry inherent in KLD, ensuring a more accurate assessment of policy inconsistency. Consequently, both qualitative and quantitative analyses led us to select JSD as our preferred metric for measuring policy inconsistency.

**Next**, we evaluate how our method's performance changes with different client importance assessment methods, comparing two alternative importance calculation approaches: **Method II** and **Method III**, with the approach used in this paper (see Eq. 24, referred to as **Method I**). The detailed mathematical expressions for Method II and Method

*Table 4.* Comparison under different FL configurations (Mean $\pm$ Standard deviation).

| | Federation with varying proportions of medium participants | | | |
|---|---|---|---|---|
| Methods | FORL (25% medium) | FDRLDR (25%) | FORL (75% medium) | FDRLDR (75%) |
| Baseline | $62.94 \pm 13.33$ | $59.23 \pm 16.62$ | $41.86 \pm 5.99$ | $45.25 \pm 4.37$ |
| **Ours + baseline** | $\mathbf{63.92} \pm 15.47$ | $\mathbf{64.71} \pm 13.90$ | $\mathbf{43.59} \pm 6.93$ | $\mathbf{45.87} \pm 3.31$ |
| | Different numbers of local training epochs | | | |
| Methods | FORL (10) | FDRLDR (10) | FORL (30) | FDRLDR (30) |
| Baseline | $38.28 \pm 10.13$ | $41.17 \pm 6.50$ | $41.46 \pm 7.11$ | $43.29 \pm 7.94$ |
| **Ours + baseline** | $\mathbf{40.26} \pm 7.27$ | $\mathbf{44.68} \pm 5.31$ | $\mathbf{46.10} \pm 9.24$ | $\mathbf{48.04} \pm 6.56$ |
| | More clients with the fixed proportion of aggregation participants | | | |
| Methods | FDRLDR (15:30) | FORL (15:30) | FDRLDR (20:40) | FORL (20:40) |
| Baseline | $47.79 \pm 6.29$ | $38.72 \pm 9.49$ | $51.05 \pm 5.57$ | $46.45 \pm 7.58$ |
| **Ours + baseline** | $\mathbf{50.80} \pm 6.42$ | $\mathbf{49.41} \pm 9.40$ | $\mathbf{52.78} \pm 8.41$ | $\mathbf{52.93} \pm 4.44$ |
| | Different proportions of aggregation participants | | | |
| Methods | FDRLDR (5:20) | FORL (5:20) | FDRLDR (15:20) | FORL (15:20) |
| Baseline | $48.67 \pm 10.02$ | $45.13 \pm 10.39$ | $43.88 \pm 6.32$ | $42.98 \pm 7.58$ |
| **Ours + baseline** | $\mathbf{51.57} \pm 6.53$ | $\mathbf{48.52} \pm 4.36$ | $\mathbf{48.62} \pm 3.69$ | $\mathbf{45.69} \pm 8.75$ |
| | Different client dataset sizes | | | |
| Methods | FDRLDR (2500) | FORL (2500) | FDRLDR (10000) | FORL (10000) |
| Baseline | $28.90 \pm 8.14$ | $28.04 \pm 7.77$ | $79.81 \pm 7.39$ | $78.18 \pm 10.67$ |
| **Ours + baseline** | $\mathbf{33.44} \pm 9.92$ | $\mathbf{31.30} \pm 8.76$ | $\mathbf{83.88} \pm 7.91$ | $\mathbf{82.45} \pm 8.78$ |
| | Introduction of random datasets | | | |
| Methods | FDQL | FDRLDR | FORL | FEDORA |
| Baseline | $34.82 \pm 10.06$ | $41.97 \pm 9.23$ | $33.10 \pm 10.89$ | $13.05 \pm 11.76$ |
| **Ours + baseline** | $\mathbf{35.35} \pm 5.60$ | $\mathbf{43.76} \pm 11.00$ | $\mathbf{39.85} \pm 11.72$ | $\mathbf{35.13} \pm 9.36$ |

III can be found in the supplementary materials. The experimental results show that using different client importance assessment methods results in varying performance for our method. When employing the method outlined in Eq. 24, we achieved the best performance. This method is also straightforward to implement, which is why we adopted it in this study. In contrast, other methods, such as $\kappa_i Q_{\theta^{Q_i}}(s, a) \times \frac{1}{Dis(\theta_i^\mu(s), a_i)}$ (Method II), are more sensitive to variations in certain factors. Even minor changes in $Dis(\theta_i^\mu(s), a_i)$ can lead to significant fluctuations in $\kappa_i Q_{\theta^{Q_i}}(s, a) \times \frac{1}{Dis(\theta_i^\mu(s), a_i)}$. In contrast, in Eq. 24, the influences of the two factors are similar.

### 5.5. Comparison under Different FL Configurations

This section compares our method with existing approaches across six different FL configurations to comprehensively demonstrate its superiority. We use FORL and FDRLDR, the two most recent methods, as baselines, with HalfCheetah as the validation task. The experimental results for the six FL configurations are summarized in Table 4, showing the final D4RL scores. These results indicate that, even with varying configurations, such as a reduced proportion of clients utilizing the expert dataset or the introduction of random datasets, as shown in Table 4, our method consistently improves SOTA offline FDRL methods, further validating its effectiveness. Despite these variations, our method bene-

fits existing approaches in both global aggregation and local training designs. First, incorporating policy inconsistency and Q-values enables a more comprehensive evaluation of client significance, enhancing global aggregation. Second, reducing the interference of a weak global model on local models contributes to improving local training.

## 6. Conclusion

This work introduces a novel FL framework that can be seamlessly integrated into existing offline FDRL approaches to enhance their performance. Specifically, we consider both policy inconsistency and Q-value to compute the weights for each client model. These weighted models are aggregated into a global model, which is then distributed to clients. During local updates, we reduce the impact of the global model on the local model when the client's model outperforms the global model. Experiments on the D4RL dataset show that our method improves seven SOTA offline FDRL methods in both return and D4RL score. Future work will explore several directions: first, hyperparameter tuning strategies, such as Population-Based Training (PBT), to automate the optimization of hyperparameters within our method; second, deployment in practical applications like autonomous driving by addressing device and model heterogeneity; and third, mitigation of various malicious attacks, such as data poisoning, that can adversely affect client models.

## Acknowledgements

The part is supported by a grant from Hong Kong Research Grant Council under General Research Fund (GRF) 11219624, the National Natural Science Foundation of China under Grant 62406067, and the Natural Science Foundation of Jiangsu Province of China under Grant BK20241298.

## Impact Statement

This paper presents work whose goal is to advance the field of offline FDRL. Offline FDRL is a pivotal technology for privacy-preserving policy learning in scenarios where raw data sharing is restricted, including energy systems, autonomous driving, and personalized medicine. In these domains, distributed agents rely on static offline datasets to learn policies, requiring effective knowledge aggregation without compromising data privacy while balancing policy performance and data fitting. Our research provides a generic, performance-enhancing framework for offline FDRL, fostering more reliable, privacy-compliant, and efficient intelligent systems across key industrial and technological sectors.

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

# A. Appendix

## A.1. Complexity Analysis

This section demonstrates the complexity introduced by the unique components of our method compared to existing offline FDRL methods. **Time complexity:** In the global aggregation phase, each client computes the Q-values and policy inconsistency for its local model, resulting in a time complexity of $O(N_a \cdot |\mathcal{D}_{0i}|)$. Then, computing each client's weight incurs a complexity of $O(N_a)$. Thus, the overall time complexity for global aggregation is $O(N_a + N_a \cdot |\mathcal{D}_{0i}|)$. In the local training phase, the time complexity for each client to compute the decay factor is $O(1)$, resulting in a total time complexity of $O(N_a)$ for $N_a$ clients. Therefore, the total time complexity is $O(N_a + N_a \cdot |\mathcal{D}_{0i}|)$, which introduces minimal additional computational cost to the existing offline FDRL methods. Lastly, since our method does not introduce new model components or alter the process of model upload and download, there is no increase in **Space complexity** or **Communication cost** compared to existing offline FDRL methods.

## A.2. The Pseudocode

The training for the server is outlined in Algorithm 1.

---

**Algorithm 1** Training procedure of our method (Server)

---

1: Initialize the global model $(\theta^\mu_{global}, \theta^Q_{global})$
2: Initialize the maximum number of communication rounds $E_c$ and the number of clients $N_a$.
3: Set $round = 1$
4: **repeat**
5:     **for** $k = 1$ to $N_a$ **do**
6:         Client $k$ receives the global model $(\theta^\mu_{global}, \theta^Q_{global})$
7:         Client $k$ updates its local model based on the received global model $(\theta^\mu_{global}, \theta^Q_{global})$
8:         Client $k$ computes its importance
9:     **end for**
10:     Compute client weights
11:     Aggregate the models from the clients
12:     Distribute the updated global model $(\theta^\mu_{global}, \theta^Q_{global})$
13:     $round + +$
14: **until** $E_c - round$

---

The training process for each client is detailed in Algorithm 2.

## A.3. Theoretical Analysis

This section provides a theoretical analysis of how our method improves upon existing offline FDRL approaches by additionally considering policy inconsistency. We start by introducing an assumption and four lemmas to prove Theorem 1. From Theorem 1, we derive Theorem 2, which demonstrates why our method improves current approaches by incorporating policy inconsistency into global aggregation. Building on Theorem 2, we establish Theorem 3, which derives the upper bound on the performance gap between our method and the theoretically optimal global policy. Finally, using Definition 1, we derive Lemma 5, leading to Theorem 4, which highlights the difference between the Q-values of our method and the optimal global policy.

**Assumption 1**: In DRL, the reward magnitude $|r(s)|$ is limited by a constant $R_M$, such that $|r(s)| \leq R_M$.

**Definition 1**: A function is Lipschitz continuous if its rate of change remains within a fixed bound. For any two points $x$ and $y$, a constant $K$ exists such that:

$$\|f(x) - f(y)\| \leq K\|x - y\| \tag{10}$$

This implies that the function's rate of change is constrained by $K$ across the entire domain.

**Lemma 1**: The research presented in (Xiong et al., 2022) indicates that the objective for DRL, $J(\pi(s))$, can be restructured

---

**Algorithm 2** Training procedure of our method (Client)

---

1: Initialize the actor $\theta^\mu$, the critics $\theta^{Q_1}$ and $\theta^{Q_2}$, along with the target networks $\theta^{\mu'}$, $\theta^{Q'_1}$, and $\theta^{Q'_2}$.
2: Initialize the dataset $\mathcal{D}$, the maximum number of communication rounds $E_c$, and the maximum duration $T$ for each local training.
3: $round = 1$
4: **repeat**
5:     Receive the global model
6:     Calculate the capability of the global model $I_{glo}$ and the capability of the local model $I$
7:     Decay $\beta$ if $I_{glo}$ is weaker than $I$
8:     **for** $t = 1$ to $T$ **do**
9:         Sample a random mini-batch $\mathcal{D}_0$ from $\mathcal{D}$ to update the model
10:         Update the critics
11:         Update the actor
12:         Every $d$ steps, update the target networks: $\theta^{Q'_j} \leftarrow \tau\theta^{Q_j} + (1-\tau)\theta^{Q'_j}$, $\theta^{\mu'} \leftarrow \tau\theta^\mu + (1-\tau)\theta^{\mu'}$, for $j = 1, 2$
13:     **end for**
14:     Each client uploads the models $\theta^\mu$, $\theta^{Q_1}$, and $\theta^{Q_2}$ to the server
15:     $round + +$
16: **until** $E_c - round$

---

as follows:

$$J(\pi(s)) = \frac{1}{1-\gamma}\mathbb{E}_{s\sim D^{\pi(s)}}[r(s)] \tag{11}$$

Here, $\frac{1}{1-\gamma}$ represents a constant positive value applicable to any given policy.

**Lemma 2**: Let $D^{\pi(s)} : \mathcal{S} \to \mathbb{R}$ be the occupancy measure associated with policy $\pi$ (Xiong et al., 2022), defined as:

$$D^{\pi(s)} = \int_{\mathcal{S}} \sum_{t=0}^{\infty}(1-\gamma)\gamma^t p_0(s)p\left(s \to s', t, \pi\right)\mathrm{d}s \tag{12}$$

Here, $p(s \to s', t, \pi)$ denotes the probability density of transitioning from state $s$ to state $s'$ in $t$ steps, under policy $\pi$. According to the findings presented in (Xiong et al., 2022), the expression for $\mathbb{E}_{s\sim D^{\pi(s)}}[r(s)]$ can be formulated as follows:

$$\mathbb{E}_{s\sim D^{\pi(s)}}[r(s)] = \int_{\mathcal{S}} r(s)D^{\pi(s)}\mathrm{d}s \tag{13}$$

**Lemma 3**: Let $f : S \subset \mathbb{R}^m \to \mathbb{R}$ be a function. If $p \leq q$ and the interval $[p, q]$ lies within $S$, then the inequality

$$\left|\int_p^q f(x)\mathrm{d}x\right| \leq \int_p^q |f(x)|\mathrm{d}x \tag{14}$$

holds true.

**Proof**: Let us start by considering the integrand $f(x)$. For each $x \in [p, q]$, we have:

$$\begin{aligned}|f(x)| \geq f(x) \quad \text{if} \quad f(x) \geq 0, \quad \text{and} \\ |f(x)| \geq -f(x) \quad \text{if} \quad f(x) < 0\end{aligned} \tag{15}$$

when we take the integral, we can directly compare the integrals of $|f(x)|$ and $f(x)$:

$$\int_p^q |f(x)|\, \mathrm{d}x \geq \int_p^q f(x)\, \mathrm{d}x \quad \text{if} \quad f(x) \geq 0,$$

$$\text{and} \quad \int_p^q |f(x)|\, \mathrm{d}x \geq - \int_p^q f(x)\, \mathrm{d}x \quad \text{if} \quad f(x) < 0 \tag{16}$$

Thus, we have:

$$\left| \int_p^q f(x)\, \mathrm{d}x \right| \leq \int_p^q |f(x)|\, \mathrm{d}x. \tag{17}$$

This completes the proof.

**Lemma 4**: The research conducted in (Xiong et al., 2022) indicates that for any two actions, $\pi_1(s)$ and $\pi_2(s)$ from the set $\mathcal{A}$, there exists a positive constant $K_\pi$ such that the following relationship is satisfied:

$$\int_{\mathcal{S}} \left| D^{\pi_1(s)} - D^{\pi_2(s)} \right| \mathrm{d}s \leq K_\pi \max_{s \in \mathcal{S}} \|\pi_1(s) - \pi_2(s)\| \tag{18}$$

**Lemma 5**: In DRL, the critic, often a neural network, computes the Q-value, and current studies assume it satisfies the Lipschitz continuity condition. From Definition 1, for a constant $K_Q$, the following inequality holds:

$$\left\| Q\left(s, \pi^A(s)\right) - Q(s, \pi^B(s)) \right\| \leq K_Q \left\| \pi^A(s) - \pi^B(s) \right\| \tag{19}$$

where $K_Q$ is a constant that quantifies the Lipschitz bound. In this context, the Lipschitz constants $K_\pi$ and $K_Q$, which are critical for Theorems 1 through 5, are standard assumptions in the theoretical analysis of reinforcement learning (Bertsekas, 2003)(Chow & Tsitsiklis, 1991)(Dufour & Prieto-Rumeau, 2013)(Dufour & Prieto-Rumeau, 2015). Additionally, both the Lipschitz constants $K_\pi$ and $K_Q$, along with occupancy measures, have been extensively employed in recent theoretical studies on deep reinforcement learning, as illustrated in study (Xiong et al., 2022). These assumptions play a vital role in enabling subsequent derivations.

**Theorem 1**: For two policies, $\pi^A(s)$ and $\pi^B(s)$, the difference between $J(\pi^A(s))$ and $J(\pi^B(s))$ can be bounded as follows:

$$
\begin{aligned}
&|J(\pi^A(s)) - J(\pi^B(s))| \\
&\overset{(Lemma 1)}{=} \frac{1}{1-\gamma} \left| \mathbb{E}_{s \sim D^{\pi^A(s)}}[r(s)] - \mathbb{E}_{s \sim D^{\pi^B(s)}}[r(s)] \right| \\
&\overset{(Lemma 2)}{=} \frac{1}{1-\gamma} \left| \int_{\mathcal{S}} r(s) \left( D^{\pi^A(s)} - D^{\pi^B(s)} \right) \mathrm{d}s \right| \\
&\overset{(Lemma 3)}{\leq} \frac{1}{1-\gamma} \int_{\mathcal{S}} |r(s)| \left| D^{\pi^A(s)} - D^{\pi^B(s)} \right| \mathrm{d}s \\
&\overset{(Assumption 1)}{\leq} \frac{R_M}{1-\gamma} \int_{\mathcal{S}} \left| D^{\pi^A(s)} - D^{\pi^B(s)} \right| \mathrm{d}s \\
&\overset{(Lemma 4)}{\leq} \frac{R_M K_\pi}{1-\gamma} \max_{s \in \mathcal{S}} \|\pi^A(s) - \pi^B(s)\|
\end{aligned}
\tag{20}
$$

This derivation shows that the difference in long-term cumulative rewards $J$ between two policies is proportional to their discrepancy, with $\frac{R_M K_\pi}{1-\gamma}$ acting as a positive constant valid for all policies.

**Theorem 2**: Let $\pi_k(s)$ be the policy of the $k$-th local model, assuming it has the minimum policy inconsistency with its corresponding offline dataset policy $\pi_k^{off}(s)$ among all clients. The global model's policy is represented as $\pi(s) = \sum_{k=1}^{N_a} w_k \pi_k(s)$, while $\pi^*(s)$ denotes the theoretical optimal global policy. The difference between the optimal policy $J(\pi^*(s))$ and the policy learned by various offline FDRL methods $J(\pi(s))$ can be bounded as follows:

$$\begin{aligned}
&|J(\pi^*(s)) - J(\pi(s))| \\
&= |J(\pi^*(s)) + J(\pi_k^{off}(s)) - J(\pi_k^{off}(s)) - J(\pi(s))| \\
&\le |J(\pi^*(s)) - J(\pi_k^{off}(s))| + |J(\pi_k^{off}(s)) - J(\pi(s))| \\
&= |J(\pi^*(s)) - J(\pi_k^{off}(s))| + |J(\pi_k^{off}(s)) + J(\pi_k(s)) - J(\pi_k(s)) - J(\pi(s))| \\
&\le |J(\pi^*(s)) - J(\pi_k^{off}(s))| + |J(\pi_k^{off}(s)) - J(\pi_k(s))| + |J(\pi_k(s)) - J(\pi(s))| \\
&\le \frac{R_M K_\pi}{1-\gamma}\left(\max_{s\in\mathcal{S}} \|\pi^*(s) - \pi_k^{off}(s)\|\right) + \frac{R_M K_\pi}{1-\gamma}\left(\max_{s\in\mathcal{S}} \|\pi_k^{off}(s) - \pi_k(s)\|\right) + \frac{R_M K_\pi}{1-\gamma}\left(\max_{s\in\mathcal{S}} \|\pi_k(s) - \pi(s)\|\right)
\end{aligned}$$
(21)

**Discussion**: In this derivation, the terms $\max_{s\in\mathcal{S}} \|\pi^*(s) - \pi_k^{off}(s)\| + \max_{s\in\mathcal{S}} \|\pi_k^{off}(s) - \pi_k(s)\|$ remain constant across different offline FDRL methods. Our approach gives greater weight $w_k$ to models from clients with minimal policy inconsistency during global aggregation, resulting in a smaller difference between $\pi(s) = \sum_{k=1}^{N_a} w_k \pi_k(s)$ and $\pi_k(s)$. Consequently, $\max_{s\in\mathcal{S}} \|\pi_k(s) - \pi(s)\|$ is reduced compared to existing offline FDRL methods. This leads to a tighter upper bound on the performance difference between our method and the theoretically optimal global policy, allowing our method to more closely approach the optimal global policy. Therefore, by incorporating policy inconsistency into global aggregation, our approach improves current offline FDRL methods.

**Theorem 3**: In offline FDRL, each client's local model minimizes the policy inconsistency with the offline dataset, bounding the difference between the offline policy and the trained local policy by $\rho_1$, such that $\max_{s\in\mathcal{S}} \|\pi_k^{off}(s) - \pi_k(s)\| \le \rho_1$. The communication frequency between global and local models is limited, with local models undergoing global aggregation after a fixed number of epochs, which bounds the difference between each local model and the global model by $\rho_2$. Additionally, since both the offline policy and the global model's optimal policy are fixed, the difference between them is bounded by $\rho_3$. Therefore, we have $\max_{s\in\mathcal{S}} \|\pi_k(s) - \pi(s)\| \le \rho_2$ and $\max_{s\in\mathcal{S}} \|\pi^*(s) - \pi_k^{off}(s)\| \le \rho_3$. As a result, the upper bound for $|J(\pi^*(s)) - J(\pi(s))|$ is given by:

$$|J(\pi^*(s)) - J(\pi(s))| \le \frac{R_M K_\pi}{1-\gamma}\left(\rho_1 + \rho_2 + \rho_3\right)$$
(22)

**Theorem 4**: Let $Q\left(s, \pi^*(s)\right)$, $Q\left(s, \pi_k(s)\right)$, $Q\left(s, \pi_k^{off}(s)\right)$, and $Q(s, \pi(s))$ denote the Q-values for the global model's optimal policy $\pi^*(s)$, the $k$-th local model's policy $\pi_k(s)$, the offline policy of the $k$-th local model $\pi_k^{off}(s)$, and the global model's policy $\pi(s)$, respectively. According to Lemma 5, the upper bound on the difference between $Q\left(s, \pi^*(s)\right)$ and $Q(s, \pi(s))$ is given by:

$$\begin{aligned}
&\|Q\left(s, \pi^*(s)\right) - Q\left(s, \pi(s)\right)\| \\
&= \left\|Q\left(s, \pi^*(s)\right) - Q(s, \pi_k^{off}(s)) + Q(s, \pi_k^{off}(s)) - Q\left(s, \pi(s)\right)\right\| \\
&\le \left\|Q\left(s, \pi^*(s)\right) - Q(s, \pi_k^{off}(s))\right\| + \left\|Q(s, \pi_k^{off}(s)) - Q\left(s, \pi(s)\right)\right\| \\
&= \left\|Q(s, \pi^*(s)) - Q(s, \pi_k^{off}(s))\right\| + \left\|Q(s, \pi_k^{off}(s)) + Q(s, \pi_k(s)) - Q(s, \pi_k(s)) - Q(s, \pi(s))\right\| \\
&\le \left\|Q(s, \pi^*(s)) - Q(s, \pi_k^{off}(s))\right\| + \left\|Q(s, \pi_k^{off}(s)) - Q(s, \pi_k(s))\right\| + \|Q(s, \pi_k(s)) - Q(s, \pi(s))\| \\
&\overset{(Lemma5)}{\le} K_Q\left(\left\|\pi^*(s) - \pi_k^{off}(s)\right\|\right) + K_Q\left(\left\|\pi_k^{off}(s) - \pi_k(s)\right\|\right) + K_Q\left(\|\pi_k(s) - \pi(s)\|\right) \\
&\le K_Q\left(\rho_1 + \rho_2 + \rho_3\right)
\end{aligned}$$
(23)

### A.4. Implementation Details

Each client uses the same DRL model, which consists of a three-layer Multi-Layer Perceptron (MLP) with 256 neurons in each hidden layer and ReLU activation for both the actor and critic. The actor's output layer uses Tanh activation. Both components share a fixed learning rate of 0.001 and are optimized using the Adam optimizer. Mini-batches contain 256 samples, with a discount factor of 0.99. The target network is updated every 2 steps using a soft update rate of 0.005. We

assess policy inconsistency using the JSD method, which our experiments have shown to be the most effective approach. $\zeta$ is set to 0.99.

## A.5. Different Methods for Calculating Client Importance

Let $Q_{\theta^{Q_i}}(s, a)$ represent the Q-value for the $i$-th client, typically generated by the first critic in the local offline DRL model, $\theta_i^\mu$ the actor for the $i$-th client, and $a_i$ the action from the $i$-th client's mini-batch $\mathcal{D}_{0i}$, which is sampled from the static dataset $\mathcal{D}_i$. The importance assessment for each client takes into account both the Q-value and the policy inconsistency $Dis(\theta_i^\mu(s), a_i)$. The importance $I_i$ can be calculated using any combination of Q-value and $Dis(\theta_i^\mu(s), a_i)$, such as Q-value minus $Dis(\theta_i^\mu(s), a_i)$. The guiding principle is that a smaller $Dis(\theta_i^\mu(s), a_i)$ is preferable, while a larger Q-value is desirable.

**Method I**: This method calculates the importance $I_i$ for the $i$-th client as follows:

$$I_i = \mathbb{E}_{(s,a)\sim\mathcal{D}_{0i}} \left[ \kappa_i Q_{\theta^{Q_i}}(s, a) - Dis(\theta_i^\mu(s), a_i) \right] \tag{24}$$

Here, $Dis(\theta_i^\mu(s), a_i)$ can be computed using any distributional difference metric, such as the squared difference: $Dis(\theta_i^\mu(s), a_i) = (\pi_{\theta_i^\mu}(s) - a_i)^2$.

**Method II**: This method fuses the Q-value and the reciprocal of policy inconsistency via multiplication to calculate the importance $I_i$ for the $i$-th client. The calculation is as follows:

$$I_i = \mathbb{E}_{(s,a)\sim\mathcal{D}_{0i}} \left[ \kappa_i Q_{\theta^{Q_i}}(s, a) \times \frac{1}{Dis(\theta_i^\mu(s), a_i) + \epsilon} \right] \tag{25}$$

Here, $\epsilon = 10^{-6}$ is a small positive constant added to the denominator to avoid division by zero, and $Dis(\theta_i^\mu(s), a_i) = (\pi_{\theta_i^\mu}(s) - a_i)^2$ (squared difference) as the policy inconsistency metric. The reciprocal of the policy inconsistency ensures that a smaller $Dis(\theta_i^\mu(s), a_i)$ leads to a larger contribution from this term, which is consistent with the guiding principle.

**Method III**: This method fuses the Q-value and the reciprocal of policy inconsistency via addition to calculate the importance $I_i$ for the $i$-th client. The calculation is as follows:

$$I_i = \mathbb{E}_{(s,a)\sim\mathcal{D}_{0i}} \left[ \kappa_i Q_{\theta^{Q_i}}(s, a) + \frac{1}{Dis(\theta_i^\mu(s), a_i) + \epsilon} \right] \tag{26}$$

Here, $\epsilon = 10^{-6}$ is a small positive constant added to the denominator to avoid division by zero, and $Dis(\theta_i^\mu(s), a_i) = (\pi_{\theta_i^\mu}(s) - a_i)^2$ (squared difference) as the policy inconsistency metric. Both the Q-value term and the reciprocal of the policy inconsistency term contribute positively to the importance $I_i$, aligning with the goal of pursuing larger Q-values and smaller policy inconsistency.

## A.6. Comparison with SOTA Offline FDRL Methods

This section presents our improvements to four SOTA methods across four MuJoCo tasks in D4RL: HalfCheetah, Hopper, Walker2d, and Ant. We evaluate these methods based on episode return, final episode return, and D4RL score. The final episode return is provided in Table 5.

Fig. 2 to 3 present the experimental results in terms of episode return, while Fig. 4 to 5 present the results based on the D4RL score. The results indicate that after integrating our method, which replaces the existing global aggregation methods, the performance of offline FDRL methods improves in almost all cases, with higher episode returns and D4RL scores. This demonstrates that our global aggregation method helps existing methods achieve better performance.

## A.7. Comparison under Different FL Configurations

This section compares our method with existing approaches across six different FL configurations to further demonstrate its superiority. **First**, we explore federation with varying proportions of medium client participants, adjusting the proportion of clients using the medium dataset to 25% and 75%. **Second**, we examine different numbers of local training epochs by changing the local training duration after each global aggregation to 10 and 30 epochs. **Third**, we maintain a fixed

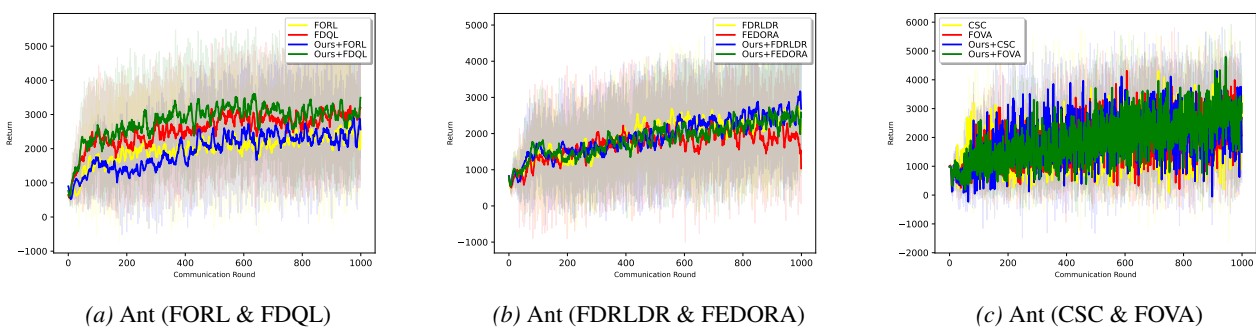

*(a)* HalfCheetah (FORL & FDQL)  *(b)* Hopper (FORL & FDQL)  *(c)* Walker2d (FORL & FDQL)

*(d)* HalfCheetah (FDRLDR & FEDORA)  *(e)* Hopper (FDRLDR & FEDORA)  *(f)* Walker2d (FDRLDR & FEDORA)

*(g)* HalfCheetah (CSC & FOVA)  *(h)* Hopper (CSC & FOVA)  *(i)* Walker2d (CSC & FOVA)

*Figure 2.* Comparison with SOTA offline FDRL methods in terms of episode return. The x-axis represents the communication rounds, and the y-axis shows the return achieved by the server in each round. The bold curve represents the average performance, while the shaded area illustrates the standard deviation across five runs with different random seeds.

*(a)* Ant (FORL & FDQL)  *(b)* Ant (FDRLDR & FEDORA)  *(c)* Ant (CSC & FOVA)

*Figure 3.* Continued: Episode return comparison on Ant environment.

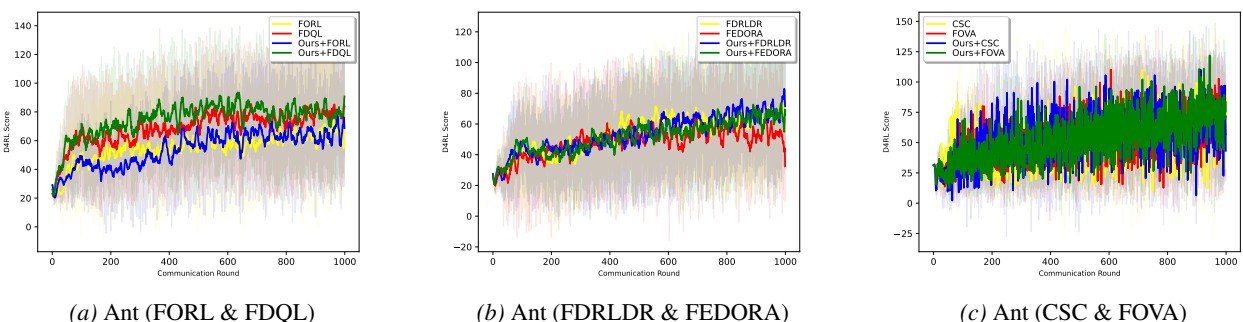

*(a)* HalfCheetah (FORL & FDQL)    *(b)* Hopper (FORL & FDQL)    *(c)* Walker2d (FORL & FDQL)

*(d)* HalfCheetah (FDRLDR & FEDORA)    *(e)* Hopper (FDRLDR & FEDORA)    *(f)* Walker2d (FDRLDR & FEDORA)

*(g)* HalfCheetah (CSC & FOVA)    *(h)* Hopper (CSC & FOVA)    *(i)* Walker2d (CSC & FOVA)

*Figure 4.* Comparison with SOTA offline FDRL methods in terms of D4RL score.

*(a)* Ant (FORL & FDQL)    *(b)* Ant (FDRLDR & FEDORA)    *(c)* Ant (CSC & FOVA)

*Figure 5.* Continued: D4RL score comparison on Ant environment.

*Table 5.* Comparison with SOTA methods in terms of final episode return (Mean $\pm$ Standard deviation).

| Methods | HalfCheetah | Hopper |
|---|---|---|
| FDQL | $5031.07 \pm 775.86$ | $1424.06 \pm 305.45$ |
| **Ours+FDQL** | $\mathbf{5747.2} \pm 1180.18$ | $\mathbf{1621.01} \pm 442.34$ |
| FDRLDR | $5716.45 \pm 534.46$ | $1426.13 \pm 476.2$ |
| **Ours+FDRLDR** | $\mathbf{5838.47} \pm 723.02$ | $1382.66 \pm 336.58$ |
| FORL | $4810.62 \pm 1083.0$ | $1506.8 \pm 349.45$ |
| **Ours+FORL** | $\mathbf{6143.2} \pm 1038.35$ | $\mathbf{1716.11} \pm 642.87$ |
| FEDORA | $3041.12 \pm 2279.27$ | $1588.71 \pm 381.44$ |
| **Ours+FEDORA** | $\mathbf{5755.13} \pm 701.9$ | $\mathbf{2727.99} \pm 1166.83$ |
| CSC | $4989.97 \pm 1757.79$ | $1351.52 \pm 331.83$ |
| **Ours+CSC** | $\mathbf{5477.17} \pm 958.52$ | $\mathbf{3139.08} \pm 466.98$ |
| FOVA | $5553.94 \pm 838.66$ | $1337.22 \pm 324.18$ |
| **Ours+FOVA** | $4977.53 \pm 721.73$ | $\mathbf{1559.18} \pm 348.5$ |

| Methods | Walker2d | Ant |
|---|---|---|
| FDQL | $3065.37 \pm 948.57$ | $3028.94 \pm 1453.11$ |
| **Ours+FDQL** | $\mathbf{3566.6} \pm 613.21$ | $\mathbf{3127.13} \pm 1356.95$ |
| FDRLDR | $3407.13 \pm 651.35$ | $2428.22 \pm 952.39$ |
| **Ours+FDRLDR** | $\mathbf{3467.93} \pm 700.12$ | $\mathbf{2959.29} \pm 1029.58$ |
| FORL | $2895.01 \pm 618.22$ | $2163.45 \pm 1311.0$ |
| **Ours+FORL** | $\mathbf{3292.47} \pm 918.16$ | $\mathbf{2667.41} \pm 1125.41$ |
| FEDORA | $3122.96 \pm 840.22$ | $1537.28 \pm 931.17$ |
| **Ours+FEDORA** | $\mathbf{5018.6} \pm 16.67$ | $\mathbf{2267.79} \pm 1212.57$ |
| CSC | $4985.43 \pm 37.62$ | $2306.14 \pm 666.26$ |
| **Ours+CSC** | $2990.89 \pm 703.58$ | $\mathbf{2498.86} \pm 1140.7$ |
| FOVA | $2744.36 \pm 515.4$ | $2397.32 \pm 1093.51$ |
| **Ours+FOVA** | $\mathbf{3280.43} \pm 1119.58$ | $\mathbf{2775.69} \pm 1334.67$ |

proportion of 50% aggregation participating clients while varying both the total number of clients and those participating in global aggregation to 30:15 and 40:20. **Fourth**, we compare the performance of different methods with client local datasets of sizes $|\mathcal{D}_i|$ set to 2500 and 10000. **Fifth**, we analyze different proportions of aggregation participants, keeping the total number of clients at 20 and setting participation ratios to 5:20 and 15:20. **Sixth**, we set up 30 clients, with 15 participating in FL; 10 clients have the expert dataset, 10 have the medium dataset, and another 10 have a random dataset, allowing us to analyze the impact of introducing lower-quality datasets on our method.

The experimental results for the six FL configurations are summarized in Fig. 6 to 17, showing the D4RL scores. These results demonstrate that, even with varying configurations, such as the reduced proportion of clients utilizing the expert dataset, our method consistently improves SOTA methods, further validating its effectiveness.

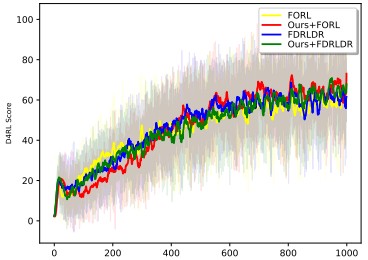

*Figure 6.* Federation with 15% medium participants

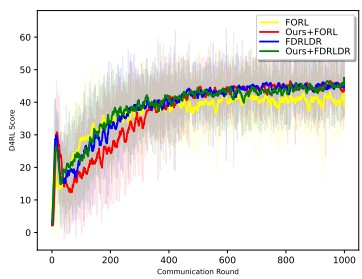

*Figure 7.* Federation with 75% medium participants

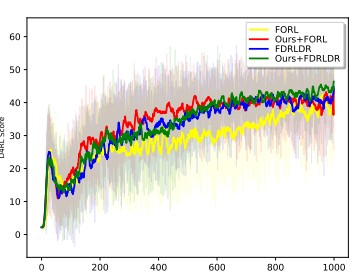

*Figure 8.* Local training with 10 epochs

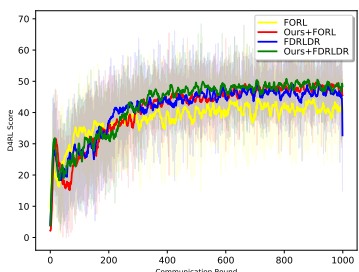

*Figure 9.* Local training with 30 epochs

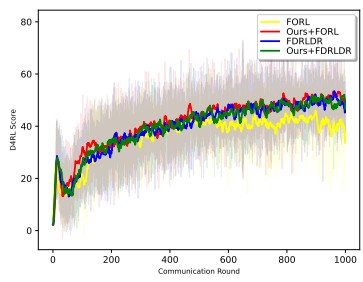

*Figure 10.* Federated learning with 30 total clients and 15 aggregation participants (fixed proportion)

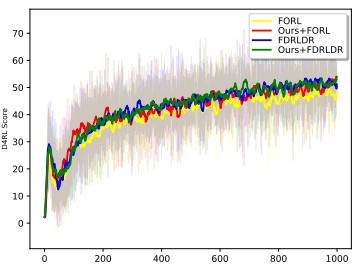

*Figure 11.* Federated learning with 40 total clients and 20 aggregation participants (fixed proportion)

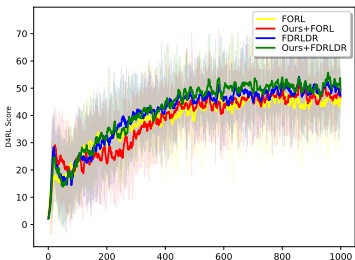

*Figure 12.* Federated learning with 20 total clients and 5 aggregation participants (different proportion)

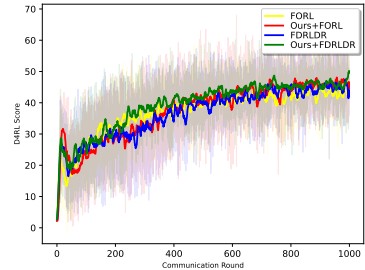

*Figure 13.* Federated learning with 20 total clients and 15 aggregation participants (different proportion)

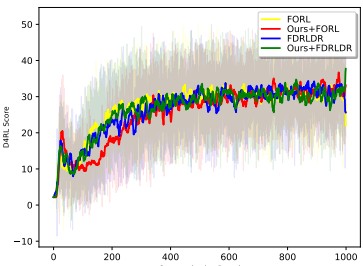

*Figure 14.* Federated learning with client dataset size of 2500 samples

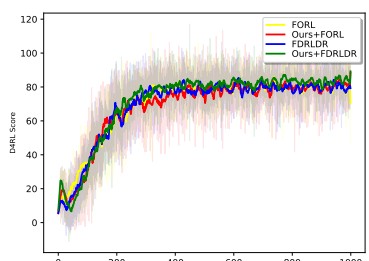

*Figure 15.* Federated learning with client dataset size of 10000 samples

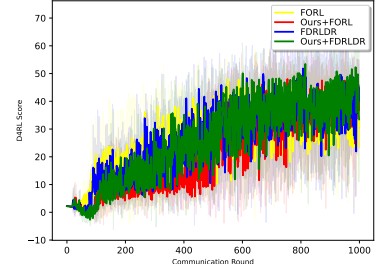

*Figure 16.* Comparison of improvements in FDQL and FEDORA with the introduction of a random dataset.

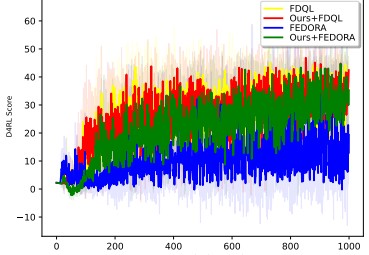

*Figure 17.* Comparison of improvements in FORL and FDRLDR with the introduction of a random dataset.

