# OpenReview forum: "Expected Returns and Policy Inconsistency-Aware Offline Federated Deep Reinforcement Learning"
_ICML.cc/2026/Conference — ICML 2026 regular_

### Official Review · Reviewer_aSW5 · 2026-03-10

**Soundness:** 3
**Presentation:** 2
**Significance:** 2
**Originality:** 3
**Overall Recommendation:** 4
**Confidence:** 2

**Summary:**

This paper introduces D4RL to improve the performance of FDRL by (1)  considering both the Q-value and the policy inconsistency when assigning weights to clients, and (2) reducing the weights of local updates for clients with higher local Q value. Numerical results show performance gain of the proposed method compare to baselines.

**Compliance With Llm Reviewing Policy:**

Affirmed.

**Final Justification:**

My concerns have been resolved. However, I'm not an expert of FDRL, therefore choose to maintain my rating since it's positive.

**Key Questions For Authors:**

Could you please clarify more about the deign of local training?

**Limitations:**

The authors did not discuss the limitations.

**Strengths And Weaknesses:**

Pros:
1. The authors provide comprehensive experiments and ablation studies, showing the effectiveness of the proposed components.
2. The method design well match the motivation, and reslove the key issues of previous research.

Cons:
1. Client reweighting is not a new thing in FL. Although exisitng methods mainly not designed for RL, the existing client weighting and client sampling methods should also be discussed.
2. I'm confused about the motivation of Eq (6). As shown in introduction, such design is to "reduce the impact of a weaker global model on the local models". To my understanding, the design aims to improve the personalization performance of clients. However, the metrics only reported the global model's performance.
3. Some hyper-parameters may hard to tune, for example, the k in Eq 1.

---

> ### Author Rebuttal · Authors · 2026-03-26
>
> We appreciate the reviewer's comments. Here, we use the abbreviations Q for Questions and W for Weakness concerns.
>
> Regarding existing client weighting and client sampling methods (W1): We apologize for our previous insufficient analysis. We have analyzed client weight calculation methods in FL as follows, and unlike existing FDRL and FL methods, we propose a novel weighting approach that considers both Q-values and policy inconsistency for offline FDRL. This explanation will be added to the final version. The client weight calculations in supervised FL involve aggregating model updates based on dataset size proportions, as seen in methods like FedProx. Other approaches utilize local test performance as the reputation score of local models to determine aggregation weights, as mentioned in [1], or assess the importance of local models based on local Shapley values, as discussed in [2]. However, since FDRL and offline FDRL are not simple supervised learning tasks like FL, we need to design client weight calculation methods that specifically cater to the characteristics of DRL. Online FDRL client weight calculations include average weighting and strategy KL divergence weighting (e.g., FedKL), while offline FDRL methods involve average weighting (e.g., FORL) and Q-value-based weighting (e.g., FEDORA). In contrast to online FDRL, offline FDRL aims not only to maximize actions' Q-values but also to reduce policy inconsistency with offline data.
>
> Regarding the report metrics focusing on global model performance (W2 & Q1): We apologize for our previous insufficient explanation; our clarification is as follows, and this will be added to the final version. There is a significant difference between offline FDRL and personalized FL regarding their learning objectives, leading to different strengths of personalization needs and consequently different performance evaluation methods. In personalized FL, each client performs a supervised learning task aimed at continuously fitting local data's personalized labels, resulting in significant variations between clients' learning objectives and the global model. This strong personalization demand renders it insufficient to evaluate FL effectiveness based solely on global model performance. In contrast, each client in offline FDRL executes an offline DRL task, where the goal is not merely to fit offline data but to train an optimal policy capable of completing a task using the given offline interaction data. Under the offline FDRL setting, all clients serve the same Markov Decision Process (MDP) task (e.g., HalfCheetah). Despite differences in the offline interaction data provided, their ultimate goal remains consistent: to collaboratively train a better policy to accomplish the task. Consequently, offline FDRL does not have a strong personalization demand, and existing offline FDRL literature predominantly uses global model performance to assess learning effectiveness.
>
> Regarding the design of local training and the motivation behind Equation (6) (Q1): In offline FDRL, clients share the same pursuit as the global model: to collaboratively train a stronger global model to better accomplish a task. Thus, all clients aim to contribute to the improvement of the global model. In this context, stronger client models are more likely to produce a better global model during aggregation. Therefore, we need to ensure the quality of client models. However, since local training requires the use of the global model, and the global model's capabilities may not always exceed those of local models (with model capability assessed by our importance evaluation metric, as indicated in Equation (1)), we evaluate the abilities of both the global and local models during local training. If the global model is weaker than the local model, we reduce its influence on the local model to ensure that the quality of the local model better serves the global aggregation. Thus, we introduce a decay factor \(\beta\) in the local training process and assess its impact on global model performance.
>
> Regarding the hyperparameter $k$ in Equation (1) (W3): $k$ is dynamically computed rather than being a hyperparameter that requires manual tuning. The value of $k$ for the $i$-th client is calculated as follows:
>
> $$k_i=\frac{1}{\frac{1}{|\mathcal{D}_{0i}|} \sum_{_{(s_i, a_i) \sim \mathcal{D}_{0i}}}\left|Q\left(s_i, a_i\right)\right|}$$
>
> Here, $\mathcal{D}_{0i}$ denotes the mini-batch for the $i$-th client, and $Q(s_i, a_i)$ is calculated based on this mini-batch. We will include the above explanation in the final version.
>
> [1]Wang Y, Kantarci B. Reputation-enabled federated learning model aggregation in mobile platforms[C]//ICC 2021-IEEE international conference on communications. 2021: 1-6.
>
> [2]Tang Z, Shao F, Chen L, et al. Optimizing federated learning on non-IID data using local Shapley value[C]//CAAI International Conference on Artificial Intelligence. 2021: 164-175.

---

> > ### Author Rebuttal · Reviewer_aSW5 · 2026-04-02
> >
> > Thank you for the response. My concerns have been resolved. However, I'm not an expert of FDRL, therefore choose to maintain my rating since it's positive.

---

### Official Review · Reviewer_KVW7 · 2026-03-12

**Soundness:** 3
**Presentation:** 4
**Significance:** 3
**Originality:** 4
**Overall Recommendation:** 5
**Confidence:** 5

**Summary:**

The authors propose a novel FL framework designed to be seamlessly integrated into existing offline FDRL approaches to enhance their performance. Existing methods typically assign client weights during global aggregation using either simple averaging or Q-values, neglecting the combined consideration of Q-values and policy inconsistency. Consequently, clients with poor data-fitting abilities but no significant expected return advantage can disproportionately degrade the global model. To solve this, the proposed method determines client weights by combining policy inconsistency, which is the distributional discrepancy between the learned policy and the offline data, and Q-values. Furthermore, the Q-values are adjusted by a scaling factor to prevent significant numerical discrepancies with the policy inconsistency metrics. Finally, if a local model outperforms the global model, a decay factor is applied to reduce the global model's influence on the local updates. Extensive experiments on D4RL datasets demonstrate that this approach improves six state-of-the-art FDRL methods.

**Compliance With Llm Reviewing Policy:**

Affirmed.

**Final Justification:**

I had an in-depth discussion with the authors, and they addressed my questions thoroughly. I recognize the contribution and value of the paper and raise my score to 5.

**Key Questions For Authors:**

Q1. How do the authors intend to rectify the formatting issues, specifically the misalignment in Table 4 and inconsistencies in abbreviation definitions and formula punctuation, to ensure compliance with ICML publication standards?

Q2. Given the importance of the theoretical foundation to the proposed FDRL algorithm, do the authors plan to integrate core proof logic or key lemmas into the main body to enhance the narrative's depth?

Q3. Have the authors considered an adaptive decay mechanism that dynamically adjusts this ratio based on the real-time performance gap between the global and local models?

Q4. Can the authors provide comparative data against these general FL methods on the same D4RL tasks?

Q5. Can the authors report specific empirical computational metrics, such as the additional training time per round, peak memory usage, or FLOPs incurred by this mechanism?

Q6. Can the authors explicitly list core assumptions as independent "Assumptions" in the main text and discuss their validity within complex, high-dimensional continuous action spaces common in D4RL?

I will adjust my score based on the authors' responses to the above questions.

**Limitations:**

yes

**Strengths And Weaknesses:**

Strengths
1. By allowing superior local models to distrust a weak global model, the framework effectively handles the "negative transfer" problem common in heterogeneous federated environments.
2. The framework is designed as a modular "wrapper" rather than a standalone algorithm. Its ability to enhance six different SOTA offline FDRL algorithms (e.g., FDQL, FEDORA, FORL) without altering their core logic demonstrates immense practical and engineering value.
3. The authors conducted experiments across a wide array of D4RL tasks and baseline methods. The inclusion of experiments with "random datasets" (noise injection) in the appendix proves the system's ability to identify and down-weight low-quality clients, showcasing its robustness in adversarial or low-data-quality scenarios.
4. The theoretical derivation in Appendix A.4, which decomposes the total offline RL error into approximation error and distributional shift error, provides a solid mathematical justification for the dual-metric (Q + Inconsistency) weighting scheme.

Weaknesses
1. The manuscript requires significant polishing to meet professional academic standards. Several errors detract from the quality of the work: 1) Table 4 contains obvious typesetting errors where merged cell contents are misaligned. 2) Standard academic practice requires defining abbreviations at their first mention in the main text (e.g., "Deep Reinforcement Learning (DRL)"), regardless of whether they were defined in the abstract. 3) Formula punctuation is inconsistent throughout the text. For instance, Equations 14 and 15 conclude with periods as part of the sentence structure, whereas Equation 1 lacks any punctuation.
2. Over-reliance on the Appendix: While the appendix serves as supplementary material, the authors have relegated all theoretical proofs to the back. This choice hampers the reading experience and obscures the logical foundation of the methodology.
3. The decay factor $\zeta$ exhibits high sensitivity, with ablation studies showing an exceptionally narrow optimal range. If $\zeta$ is not tuned perfectly, the mechanism risks exacerbating negative transfer.
4. Absence of Standard FL Baselines: The study focuses exclusively on FDRL-specific methods. While I acknowledge the authors' argument that general FL methods are not inherently designed for RL, the inclusion of baselines like MOON is essential.
5. Although theoretical complexity is discussed, the paper would be significantly strengthened by reporting specific empirical metrics (e.g., training time per round, memory overhead, or FLOPs) to quantify the cost.
6. Definition 1 in the appendix appears to be a standard concept but lacks a citation to the original literature.
7. Furthermore, critical assumptions, such as Lipschitz continuity, are currently buried within the lemmas. For better clarity and rigor, these should be explicitly stated as independent "Assumptions" and justified within the context of offline RL.

---

> ### Author Rebuttal · Authors · 2026-03-27
>
> We appreciate the reviewers' comments. In what follows, we use the abbreviations Q for Questions, W for Weakness concern.
>
> Regarding W1 & Q1, we will adjust the table formatting to ensure that merged cell contents are properly aligned and check for consistency across all tables. Full definitions for abbreviations will be provided upon their first appearance in the main text, and we will standardize punctuation in all formulas. Additionally, we will review other formatting aspects and rigorously proofread the entire document according to ICML publishing standards to ensure compliance.
>
> Regarding W2 & Q2, we will integrate key theoretical conclusions and important lemmas, such as the assumptions in the appendix and core conclusions from Theorems 1 to 3, into the main text. This will clearly demonstrate the theoretical foundation of our method while retaining detailed mathematical derivations in the appendix.
>
> Regarding W3 and Q3, since DRL is sensitive to hyperparameters, the decay coefficient ζ is a hyperparameter of our method that also affects its performance. Therefore, we have currently chosen a value for ζ through sensitivity testing that yields good performance. The DRL community is actively exploring advanced hyperparameter tuning strategies, such as Population-Based Training (PBT), for automated optimal configurations. In the future, we will consider integrating these methods for the automatic adjustment of the decay coefficient ζ. Secondly, we have considered dynamically adjusting β based on the real-time performance gap between the global and local models, but we recognize two limitations. First, the performance of local models may change at each local training step, requiring us to frequently measure the difference from the global model and adjust the decay factor. This process could impose a significant computational burden, conflicting with the limited resources available in FDRL. Second, the inherent variability in client DRL training means that frequent adjustments (at every local training step) could exacerbate training instability. Due to these two limitations, we have not implemented such a fine-grained adaptive decay mechanism at this time.
>
> Regarding W4 and Q4, based on your suggestion, we conducted new comparisons with MOON [1], a general FL method, on the same D4RL tasks, specifically in terms of final D4RL score. This experiment followed the configuration outlined in Section 5.2 and was conducted across five random seeds. The new results will be included in the final version, as follows. The experimental results demonstrate that our method helped MOON achieve a higher D4RL score.
>
> | Method          | HalfCheetah    | Hopper         | Walker2d        | Ant             |
> |-----------------|----------------|----------------|-----------------|-----------------|
> | MOON            | 33.45 ± 5.95   | 48.40 ± 17.27  | 65.14 ± 17.46   | 15.59 ± 5.31    |
> | Ours+MOON       | 44.84 ± 10.46  | 60.24 ± 10.94  | 69.13 ± 16.42   | 80.90 ± 12.02   |
>
> Regarding W5 and Q5, as per your recommendation, we tested the average Floating Point Operations (FLOPs) and average Video Random Access Memory (VRAM) usage in MB for all clients during each local training epoch to evaluate the hardware efficiency of our method. FLOPs measure computational workload, while VRAM usage assesses memory consumption. The experimental configuration will follow Section 5.2, and these new results will also be included in the final version, as follows. Our tests indicate that the FDRLDR [2] method and Ours+FDRLDR method have similar VRAM and FLOPs, both approximately 88.99 MB and 4.11e+06, respectively, for HalfCheetah, while for Hopper, the VRAM and FLOPs are around 86.15 MB and 3.98e+06. Thus, our method does not impose a significant computational burden on the baseline methods.
>
> Regarding W6 and Q6, we will explicitly list the core assumptions, such as Lipschitz continuity, as independent "assumptions" in the main text of the final version and provide additional analysis of their validity in the context of offline DRL, as follows. In fact, the assumption that Q-values and policies satisfy Lipschitz continuity is widely adopted in DRL literature. Since we often use neural networks to parameterize the Q-value and policy in both DRL and offline DRL, the Lipschitz continuity of Q-values and policies is generally satisfied [3]. Thus, this assumption naturally holds in high-dimensional action space datasets like D4RL.
>
> [1] Li Q, He B, Song D. Model-contrastive federated learning. Proceedings of the IEEE/CVF conference on computer vision and pattern recognition. 2021: 10713-10722.
>
> [2] Rengarajan, D., Ragothaman, N., Kalathil, D., & Shakkottai, S. (2024). Federated ensemble-directed offline reinforcement learning. Advances in Neural Information Processing Systems, 37, 6154-6179.
>
> [3] Gouk H, Frank E, Pfahringer B, et al. Regularization of neural networks by enforcing Lipschitz continuity. Machine Learning, 2021, 110(2): 393-416.

---

> > ### Author Rebuttal · Reviewer_KVW7 · 2026-04-01
> >
> > Thank you for the authors’ response. I think it has addressed most of my concerns. I agree with the reply to W3 and Q3. DRL is highly sensitive to hyperparameters. Although many people still claim that PPO works well, my impression is that its performance is only average.
> >
> > In addition, I believe that FedRL is still largely at a theoretical stage, and real world deployment remains difficult. So I would like to further discuss its future development with the authors. Do you think FedRL should place more emphasis on the federated setting, or on the practical realism of RL? RL requires a large amount of data for training. In real scenarios, data from different environments, often called environmental heterogeneity, is unavoidable. Could FL truly handle this?

---

> > > ### Author Response · Authors · 2026-04-03
> > >
> > > Thank you for your comments. Below are our responses to the follow-up questions on future research in Federated Reinforcement Learning (RL).
> > >
> > > On Hyperparameters in RL: We agree that identifying optimal configurations remains an ongoing challenge in the RL community. We will actively investigate effective self-tuning strategies for RL hyperparameters in future work.
> > >
> > > On Federated Setting and Practical Realism of RL: Since the goal of federated RL is to collaboratively train multiple RL clients to obtain a powerful global model that performs well on a specific task, we believe that both the federated setting and the practical realism of RL are equally important and complementary, for the following reasons:
> > >
> > > 1) Each client in federated RL is an RL model. Therefore, it is essential to address practical RL challenges such as mitigating environmental noise in policy learning, achieving safe policy optimization in hazardous scenarios (e.g., lane changing and overtaking in autonomous driving), and enabling continual RL under limited hardware capabilities. This highlights the importance of studying practical realism in RL.
> > >
> > > 2) RL clients in federated settings need to collaborate via knowledge sharing. In the real world, clients can vary significantly (e.g., some autonomous vehicles operate on city roads while others are on highways, and their hardware capabilities may also differ). These practical factors pose challenges for the deployment of federated aggregation in federated RL, making the study of effective federated architectures both critical and actively pursued in the current federated RL community. To address this challenge, we envision a potential solution for future exploration: Traditional federated learning, such as supervised FL, has made significant progress in overcoming environmental heterogeneity. These techniques can be adapted to federated RL and integrated with its characteristics to design aggregation strategies that handle heterogeneity. For example, autonomous vehicle clients in heterogeneous environments (e.g., city roads and highways) could upload and aggregate parts of their models, such as shared representation modules, while retaining their own decision-making modules. This approach enhances representation capabilities through federated learning while preserving decision-making abilities. In this process, the model aggregation step can leverage advanced global aggregation techniques from supervised federated learning.
> > >
> > > 3) Addressing practical challenges in RL strengthens each client, which in turn contributes to a more robust global model during federated aggregation. An effective aggregation method also ensures that clients can integrate knowledge from others to improve their ability to handle real-world challenges.
> > >
> > > Finally, we thank the reviewer for the insightful discussion. We will incorporate the above points into the future research directions of the final version.

---

### Official Review · Reviewer_SmVa · 2026-03-14

**Soundness:** 3
**Presentation:** 2
**Significance:** 2
**Originality:** 3
**Overall Recommendation:** 4
**Confidence:** 3

**Summary:**

The paper proposes a generic plug-in framework to improve existing offline Federated Deep Reinforcement Learning (FDRL) algorithms. The core idea is to make client weighting during global model aggregation aware of both expected returns (via scaled Q-values) and policy inconsistency (distributional mismatch between learned policy and offline data behaviour, measured e.g. via squared error or JSD on action distributions). During local training, the influence of the downloaded global model is dynamically reduced (via a decaying multiplier β) whenever a client's own model already shows better combined performance (higher importance score I) than the current global model. The authors claim the approach is agnostic to the underlying offline FDRL algorithm and only adds very modest computation/communication overhead. Experiments are performed on D4RL MuJoCo tasks with 20 heterogeneous clients (mixture of medium & expert data), random client participation, and six recent offline FDRL baselines; reported improvements appear in normalized return and D4RL score.

**Compliance With Llm Reviewing Policy:**

Affirmed.

**Key Questions For Authors:**

(1) It remains unclear how much gain comes from 1) using both Q and inconsistency for weighting; 2) scaling the Q-value; 3) using JSD vs. simpler squared difference, and 4) the dynamic decay mechanism during local training. Without careful ablations the reader cannot judge which part is truly essential.

(2) Only D4RL MuJoCo locomotion tasks, only medium+expert mixture, only 20 clients, only one type of heterogeneity (data quality), no medium-replay / medium-expert mixtures, no very small local datasets, no non-MuJoCo domains (e.g. AntMaze, kitchen, Adroit, or Meta-World). Generalization to harder or qualitatively different offline FDRL settings is not demonstrated.

(3) Missing analysis of communication-efficiency trade-offs. While the method claims “minimal overhead”, it requires every participating client to compute and upload not only model parameters but also a scalar importance score I every round. More importantly, no experimental studies aboud 1)performance when communication is very infrequent (e.g., local training for 100–200 epochs per round); 2) impact of compressing model updates (the method assumes full model exchange); 3) behaviour under partial client participation rates much lower than 50%; 4) In real edge-based FDRL deployments, communication cost is often the dominant constraint — yet this dimension is not investigated.

(4 ) Decay rate ζ = 0.99, σ = 0.15 for JSD, exact form of I (subtract vs. divide vs. other combinations), mini-batch size for importance computation, number of local epochs per round (T=20), fraction of participating clients, etc. are all fixed without sensitivity analysis. Many FL/RL methods are known to be brittle to such choices.

(5) The provided pages do not contain any tables, figures of final performance, or variance across seeds. The reader cannot assess 1) the magnitude of improvement; 2) consistency across environments; 3) whether gains are statistically significant; 4) whether some baselines collapse on certain settings while the proposed method remains stable.

**Limitations:**

yes

**Strengths And Weaknesses:**

The Strength of this paper:

(1) Addresses two arguably under-explored and practically relevant weaknesses of current offline FDRL literature at the same time: (a) one-sided client weighting (only Q-value or uniform), (b) potentially harmful full adoption of a weak global model.

(2) The proposed importance score I that combines policy inconsistency and (scaled) Q-value is conceptually clean and aligns well with two fundamental objectives of offline RL — maximizing long-horizon return while staying close to the support of the offline data.

(3) The dynamic decay mechanism during local updates is intuitive and elegant; it automatically protects good local models from degradation without introducing new hyperparameters that need heavy tuning per task.

(4) Claims generic plug-ability into many existing offline FDRL methods and shows consistent gains across six recent baselines — if the numbers hold, this is a practical selling point.

(5) Provides both complexity analysis (showing negligible overhead) and (reportedly) a theoretical justification in the appendix, which is more than many empirical FL/RL papers offer.

The weaknesses of this paper:

Weaknesses / Major Concerns

(1) It remains unclear how much gain comes from 1) using both Q and inconsistency for weighting; 2) scaling the Q-value; 3) using JSD vs. simpler squared difference, and 4) the dynamic decay mechanism during local training. Without careful ablations the reader cannot judge which part is truly essential.

(2) Only D4RL MuJoCo locomotion tasks, only medium+expert mixture, only 20 clients, only one type of heterogeneity (data quality), no medium-replay / medium-expert mixtures, no very small local datasets, no non-MuJoCo domains (e.g. AntMaze, kitchen, Adroit, or Meta-World). Generalization to harder or qualitatively different offline FDRL settings is not demonstrated.

(3) Missing analysis of communication-efficiency trade-offs. While the method claims “minimal overhead”, it requires every participating client to compute and upload not only model parameters but also a scalar importance score I every round. More importantly, no experimental studies aboud 1)performance when communication is very infrequent (e.g., local training for 100–200 epochs per round); 2) impact of compressing model updates (the method assumes full model exchange); 3) behaviour under partial client participation rates much lower than 50%; 4) In real edge-based FDRL deployments, communication cost is often the dominant constraint — yet this dimension is not investigated.

(4 ) Decay rate ζ = 0.99, σ = 0.15 for JSD, exact form of I (subtract vs. divide vs. other combinations), mini-batch size for importance computation, number of local epochs per round (T=20), fraction of participating clients, etc. are all fixed without sensitivity analysis. Many FL/RL methods are known to be brittle to such choices.

(5) The provided pages do not contain any tables, figures of final performance, or variance across seeds. The reader cannot assess 1) the magnitude of improvement; 2) consistency across environments; 3) whether gains are statistically significant; 4) whether some baselines collapse on certain settings while the proposed method remains stable.

---

> ### Author Rebuttal · Authors · 2026-03-25
>
> We appreciate the reviewer's comments. Here, we use Q for Questions and W for Weakness concerns.
>
> On Ablation (W1 & Q1): Firstly, using both Q-values and inconsistency for weighting is crucial for our global aggregation and local training, as it offers a shared method for assessing model capability. Without this design, our method's global aggregation and local training cannot be performed. Furthermore, employing only Q or inconsistency to evaluate model performance has been demonstrated to be suboptimal in Section 5.3. Secondly, scaling the Q-value ensures its numerical magnitude is comparable to that of inconsistency since the Q-value's scale is generally much higher than that of inconsistency. If not scaled, it would revert to a Q-value-based aggregation, akin to the baseline method FEDORA, which our method surpasses. Thirdly, Section 5.4 highlighted performance gains from using JSD over Squared Difference, emphasizing the superiority of JSD. Finally, in local training, our method differs from the baseline by introducing a dynamic decay mechanism (with an initial decay factor of 1). Section 5.3 showed that removing this mechanism (i.e., OWD + baseline) resulted in inferior performance compared to the full version of our method, underscoring the necessity of the dynamic decay mechanism.
>
> On Diverse FDRL Configurations (W2 & Q2 & W3 & Q3 & W4 & Q4): Section 5.5 evaluates our method under various FL configurations, as follows. Firstly, we evaluated the impact of increasing the number of clients to 30 and 40. Secondly, we tested with expert-medium-random datasets, which are more challenging than medium-replay/medium-expert, demonstrating the advantages of our method in the difficult setting. Thirdly, we assessed performance under varying local dataset sizes, including very small datasets with as few as 2,500 training samples, while typical offline DRL utilizes 1 million samples. Fourthly, we tested different client participation rates (e.g., 25%) and varying local training epochs per round. Additionally, we conducted extra tests as requested, using sparse communication frequency (communicating once every 100 local training epochs, only one-tenth of the original frequency) on HalfCheetah across five random seeds, with 10 clients using expert data and another 10 using medium data. The final D4RL scores for FORL and FDRLDR were 36.72±8.75 and 40.95±5.27, respectively, while our method combined with FORL and FDRLDR achieved 41.90±8.35 and 43.76±4.95, improving the baselines. This new result will be included in the final version. Lastly, we did not use tasks like AntMaze and Kitchen because these tasks introduce unique challenges; for example, AntMaze and Kitchen involve sparse rewards, necessitating enhanced exploration capabilities for FDRL, whereas Adroit and Meta-World focus on testing the ability to mitigate catastrophic forgetting, which is a crucial challenge for improving FDRL's generalization ability. These additional challenges deviate from the primary focus of this paper.
>
> On Sensitivity Testing (W4 & Q4): Firstly, Section 5.3 analyzed the sensitivity of the decay coefficient ζ. Secondly, Section 5.4 examined the sensitivity of various forms of the importance score I (subtraction, division, and addition). Additionally, sensitivity testing for the number of local epochs per round and the fraction of participating clients was conducted in Section 5.5. Thirdly, the hyperparameter configuration for JSD used is the commonly accepted setting, which demonstrated strong performance in our experiments; therefore, we did not conduct further sensitivity analysis. Finally, other hyperparameters, such as mini-batch size, were kept consistent with the baseline method, as they were not introduced by our approach.
>
> On Communication Efficiency (W3 & Q3): Existing offline FDRL methods require clients to upload complete models and importance scores. We have not altered this requirement; we merely proposed a new method for calculating importance. Thus, there is no additional communication overhead.
>
> On Data Presentation (W5 & Q5): Firstly, the data in each table represent the average D4RL score over the final 10 communication rounds during training from five different seeds, labeled "Mean," along with the corresponding standard deviation, labeled "Standard Deviation." Thus, the mean and standard deviation of each data represent the average final performance and std across the five seeds, respectively. Additionally, the thick curves and shaded areas in each plot indicate the averages and standard deviations of the five seeds, respectively. Secondly, each test unit utilized five random seed runs, and we conducted extensive testing (over 50 test units), demonstrating the widespread effectiveness of our method across various scenarios. Notably, when introducing random client datasets and very small client datasets (in Section 5.5), baseline methods performed poorly, while our method significantly improved their performance.

---

### Official Review · Reviewer_oQWH · 2026-03-24

**Soundness:** 3
**Presentation:** 3
**Significance:** 3
**Originality:** 3
**Overall Recommendation:** 4
**Confidence:** 3

**Summary:**

The paper proposes an adaptive method for the FDRL algorithm. Compared to existing methods that assign client weights based on averaging or Q-Values without consideration of policy inconsistency with Q-Values. The proposed method improves over the baselines when evaluated for return and D4RL score.

**Compliance With Llm Reviewing Policy:**

Affirmed.

**Key Questions For Authors:**

With weaknesses

**Limitations:**

What are the fail scenario where the method may fail? Encourage the authors to test that setting

**Strengths And Weaknesses:**

Strengths:

1) Compared to SOTA methods, the method considers both policy inconsistency and Q-values for averaging in global round.

2) The model accounts for weaker global model during aggregation

3) Theoretical analysis and empirical evaluation is provided for the proposed method

Weaknesses:

1) Certain results in the table show large variance especially Table 1. Maybe include more than 5 seeds to report true performance gains

---

> ### Author Rebuttal · Authors · 2026-03-25
>
> We appreciate the reviewer' s comments. In what follows, we will use the abbreviations L for Limitations, Q for Questions, and W for Weaknesses.
>
> Regarding W1 and Q1, we appreciate the reviewer's suggestions. We chose to use five random seeds for the following reasons, and to further validate this, we will include a comparison with at least 10 seeds in the final version, reporting both the mean and standard deviation to comprehensively demonstrate the effectiveness of our method. Firstly, using five random seeds in experiments is common practice in current DRL and FDRL literature, with the baseline methods we compared also employing 3-5 random seeds, as seen in [1][2]. To ensure a fair comparison, we adopted five random seeds as well. Secondly, although each test unit comparing our method with the baseline was conducted under five randomly selected seeds, our extensive testing (over 50 test units) mitigates the impact of any single test unit's results, demonstrating the widespread effectiveness of our approach. Finally, the DRL community emphasizes the mean return across multiple random seed runs over variance, as this reflects average performance across repeated trials. Thus, even when the variance of returns is high in some cases, a mean return higher than the baseline still indicates the method's effectiveness. Our experimental results (e.g., Table 1) show that our method improves the baseline in most cases regarding mean return.
>
> Regarding possible fail scenarios (L1), we apologize for our previous insufficient analysis. In the final version, we will analyze potential failure scenarios as future research directions, specifically as follows. In real-world applications, FDRL clients may face various malicious attacks, such as data poisoning by adversaries. When some clients participating in offline FDRL utilize poisoned datasets, these contaminated data can disrupt the original distribution of normal data, leading to model instability, failure to converge, and potentially causing policies to learn dangerous actions. This could result in existing offline FDRL methods failing. Therefore, we will focus on detecting and defending against poisoned clients in the future.
>
> [1] Yue, S., Qin, Z., Hua, X., Deng, Y., & Ren, J. (2024, May). Federated offline policy optimization with dual regularization. In IEEE INFOCOM 2024-IEEE Conference on Computer Communications (pp. 811-820). IEEE.
>
> [2] Qiao, N., Yue, S., Ren, J., & Zhang, Y. (2025). FOVA: Offline Federated Reinforcement Learning With Mixed-Quality Data. IEEE Transactions on Networking, 34, 2031-2046.

---

> > ### Author Rebuttal · Reviewer_oQWH · 2026-04-03
> >
> > Thanks for the clarification. I maintain the positive assessment of the paper.

---

### Decision · Program_Chairs · 2026-04-30

**Decision:**

Accept (regular)

**Comment:**

This paper studies offline federated deep reinforcement learning and proposes a generic framework that makes global aggregation aware of both expected returns and policy inconsistency, while also reducing the influence of a weak global model during local training. Reviewers found the problem relevant and the proposed framework practically useful, especially because it can be integrated into multiple existing offline FDRL methods.

After considering the reviews, rebuttal, and discussion, I find the paper strong enough for acceptance. Reviewers viewed the core idea as well motivated, and the empirical results showed consistent improvements across several offline FDRL baselines and federated settings. The rebuttal also addressed many of the main concerns by clarifying the motivation of the local training design, adding ablations and sensitivity analysis, and providing additional evidence on broader federated configurations and computational overhead.

The paper is not without limitations. Reviewers noted that the evaluation remains centered on D4RL MuJoCo tasks, that some aspects of the weighting design and theory could be better justified, and that the paper would benefit from clearer positioning relative to related federated weighting methods. However, these concerns do not outweigh the overall contribution.

Overall, this is a solid contribution to offline federated reinforcement learning with clear practical value.